# ADAPTING, FAST AND SLOW:
# TRANSPORTABLE CIRCUITS FOR FEW-SHOT LEARNING

## ABSTRACT

Generalization across the domains is not possible without asserting a structure that constrains the unseen target domain w.r.t. the source domain. Building on causal transportability theory, we design an algorithm for zero-shot compositional generalization which relies on access to qualitative domain knowledge in form of a causal graph for intra-domain structure and discrepancies oracle for inter-domain mechanism sharing. *Circuit-TR* learns a collection of *modules* (i.e., local predictors) from the source data, and *transport*/compose them to obtain a *circuit* for prediction in the target domain if the causal structure licenses. Furthermore, circuit transportability enables us to design a supervised domain adaptation scheme that operates without access to an explicit causal structure, and instead uses limited target data. Our theoretical results characterize classes of few-shot learnable tasks in terms of graphical circuit transportability criteria, and connects few-shot generalizability with the established notion of *circuit size complexity*; controlled simulations corroborate our theoretical results.

## 1 INTRODUCTION

Machine learning deals with generalizing patterns from finite samples to the distribution that generates these samples. The classical sample-to-population guarantees (Vapnik, 1991; 1998) rely on the assumption that *target domain*, where the solution would be evaluated, entails a data distribution identical to the *source domain*, where the training happens. However, in practice the performance would take a serious even under small qualitative differences between source and target domains. This problem is known broadly as a *distribution shift* in ML, and generalizability or external validity in a broader scientific context. In particular, the *domain generalization* task refers to a situation where the learner has access to typically large data collected from one or multiple source domains and no data from the target domain. This is an extreme case of the *domain adaptation* problem where the learner also has access to a small amount of labeled data collected from the target domain.

Theoretical understanding of generalization across domains is challenging. Arbitrary differences between the source and target domains inevitably imposes a barrier for learning, as there would be no basis for usefulness of source data in the target learning task. Thus, a formal approach to this problem necessitates establishing a notion of *structure* that specifies what the target domain can be in relation to the source domains. Then, one can imagine a carefully designed algorithm that leverages this structure and uses only the statistical associations present in the source data that would provably remain stable/invariant in the target, thus achieve a prediction rule with out-of-distribution guarantees (Peters et al., 2016; Rojas-Carulla et al., 2018; Rothenhäusler et al., 2021).

Domain adaptation in prediction tasks involving covariates $X$ and label $Y$ has been studied in the literature (Ben-David et al., 2006; Blitzer et al., 2007; Mansour et al., 2009a; Ben-David et al., 2010; Yang et al., 2010; Hanneke and Kpotufe, 2019; 2024), where various notions of divergence between the source and target $X$ distribution are used as proxies for domain-relatedness. Other work in this area leverages distributional assumptions relating source and target, e.g., Blitzer et al. (2011); Ben-David and Schuller (2003); Baxter (2000; 1997) where learning in the source yields smaller complexity for learning in the target, e.g., through learning a shared *representation*.

Humans are particularly effective in transferring knowledge across domains (Lake et al., 2017; Marcus, 2001; Tenenbaum et al., 2011), and causality is known to be the pillar of human understanding and

decision making, especially under changing circumstances (Gopnik et al., 2004; Schulz and Gopnik, 2004). Principles of generalization to the unseen from a causal perspective has been extensively studied under the rubrics of *transportability* (Pearl and Bareinboim, 2011; Bareinboim et al., 2013; Correa and Bareinboim, 2019; 2020; Jalaldoust and Bareinboim, 2024; Jalaldoust et al., 2024), and also through the lens of statistical invariances entailed by an implicit causal structure (Peters et al., 2016; Magliacane et al., 2018; Koyama and Yamaguchi, 2021; Li et al., 2018). In DA, since *some* target data is available, the learner would always have the choice of discarding the source data entirely, and relying solely on the target data. Thus, the theoretical question in DA is not whether it is possible to *learn*, but how fast learning can take place and how to best leverage the data from the source domains. In this paper, we seek to characterize the situations where certain aspects of the source data deems generalizable, thus allowing zero-shot/few-shot learning of the target (i.e., fast adaptation), and when learning from the source data hinders learning in the target (i.e., slow adaptation), and what lies in between these two extremes. Our contributions are the following:

1. **Circuit Transportability.** In Section 2, we define *module-transportability* which extends the transportability machinery to account for a domain knowledge that specifies shared modules across different positions/variables. We introduce module-TR algorithm (Algorithm 1) to decide module-transportability using a more fine-grained causal graph and richer notion of domain discrepancies that expand the traditional transportability toolbox. Furthermore, we introduce the notion of circuit-transportability, which not only accounts for shared modules across the positions and domains, but also considers *sequential composition* of the modules in form of *circuits*, to enable transport in some cases that module-TR fails. We devise circuit-TR algorithm (Algorithm 2) that operates using the causal graphs and discrepancy oracle (Definition 2.5) as a recipe for computing target predictor from the source data.

2. **Adaptation.** In Section 3 we consider a situations where such elaborate structural knowledge is not available, and instead, the learner has access to labeled target data. We introduce circuit-AD, a learning scheme that uses circuit-TR as a subroutine to compute a class of candidate predictors from the source data alone, and then uses the labeled target data to choose the best-in-class predictor. We prove a performance guarantee for circuit-AD; it achieves fast adaptation with error rate $\mathcal{O}(\sqrt{\frac{\text{poly}(T)}{n}})$ using only $n$ target samples, if the ground truth is circuit-transportable with a graph of size $\mathcal{O}(T)$. Our findings draw a natural connection between few-shot learnability and *circuit complexity* – a well-studied topic in theoretical computer science. Since the symbolic circuit-AD is computationally inefficient, in Section 4 we provide a gradient-based heuristic to approximate its solution, and show it's effectiveness in controlled synthetic experiments.

**Preliminaries.** We use capital letters to denote variables ($X$), small letters for their values ($x$), bold letters for sets of variables ($\mathbf{X}$) and their values ($\mathbf{x}$), and use caligraphic letters ($\mathcal{X}$) to denote their support. A conditional independence statement in distribution $P$ is written as $(\mathbf{X} \perp\!\!\!\perp \mathbf{Y} \mid \mathbf{Z})_P$. A $d$-separation statement in some graph $\mathcal{G}$ is written as $(\mathbf{X} \perp\!\!\!\perp_d \mathbf{Y} \mid \mathbf{Z})$. To denote $P(\mathbf{Y} = \mathbf{y} \mid \mathbf{X} = \mathbf{x})$, we use the shorthand $P(\mathbf{y} \mid \mathbf{x})$. The basic semantic framework of our analysis relies on Structural Causal Models (SCMs) (Pearl, 2009, Definition 7.1.1), which are defined below.

**Definition 1.1.** *An SCM $\mathcal{M}$ is a tuple $M = \langle \mathbf{V}, \mathbf{U}, \mathcal{F}, P \rangle$ where each observed variable $V \in \mathbf{V}$ is a deterministic function of a subset of variables $\mathbf{Pa}_V \subset \mathbf{V}$ and latent variables $\mathbf{U}_V \subset \mathbf{U}$, i.e., $v := f_V(\mathbf{pa}_V, \mathbf{u}_V), f_V \in \mathcal{F}$. The unobserved variables $\mathbf{U}$ follow a distribution $P(\mathbf{u})$.* □

We assume the model to be recursive, i.e. that there are no cyclic dependencies among the variables. SCM $\mathcal{M}$ entails a probability distribution $P^{\mathcal{M}}(\mathbf{v})$ over the set of observed variables $\mathcal{V}$ such that

$$P^{\mathcal{M}}(\mathbf{v}) = \int_{\mathcal{U}} \prod_{V \in \mathbf{V}} P^{\mathcal{M}}(v \mid \mathbf{pa}_V, \mathbf{u}_V) \cdot P(\mathbf{u}) \cdot d\mathbf{u}, \tag{1}$$

where the term $P(v \mid \mathbf{pa}_V, \mathbf{u}_V)$ corresponds to the function $f_V \in \mathcal{F}$ in the underlying structural causal model $\mathcal{M}$. It also induces a causal diagram $\mathcal{G}_{\mathcal{M}}$ in which each $V \in \mathbf{V}$ is associated with a vertex, and we draw a directed edge between two variables $V_i \to V_j$ if $V_i$ appears as an argument of $f_{V_j}$ in the SCM, and a bi-directed edge $V_i \leftrightarrow V_j$ if $\mathbf{U}_{V_i} \cap \mathbf{U}_{V_j} \neq \varnothing$ or $P(\mathbf{U}_{V_i}, \mathbf{U}_{V_j}) \neq P(\mathbf{U}_{V_i}) \cdot P(\mathbf{U}_{V_j})$, that is $V_i$ and $V_j$ are confounded (Bareinboim et al., 2022).

Throughout this paper, we only consider discrete-valued variable, and assume the observational distributions entailed by the SCMs satisfy strict positivity assumption, that is, $P^{\mathcal{M}}(\mathbf{v}) > \epsilon$, for every

**v** and a known constant $\epsilon$. We will also operate non-parametrically, i.e., making no assumption about the particular functional form or the distribution of the unobserved variables.

## 2 CIRCUIT TRANSPORTABILITY

We consider a classification problem where $\mathbf{X} = \{X_1, X_2, ..., X_M\}$, and $Y$ are discrete-valued variables that take value in a shared finite vocabulary $\mathcal{V}$. The objective is predicting the label $Y$ using covariates $\mathbf{X}$, i.e., learning $P^*(y \mid \mathbf{x}_{1:M})$. There is a loss function $\ell(\mu; y, \mathbf{x})$, and the risk is defined as the expected loss $R_{P*}(\mu) := \mathbb{E}_{P*}[\ell(\mu; Y, \mathbf{X})]$. The true risk minimizer is denoted as $\mu_* \in \arg\min_{\mu:\mathcal{V}^{M_X} \to \text{simplex}(|\mathcal{V}|^{M_Y})} R_{P*}(\mu)$, and the empirical risk minimizer w.r.t. data $D$ is denoted as,

$$\hat{P}(y \mid \mathbf{x}; D) \in \arg\min_{\mu:\mathcal{V}^{M_X} \to \text{simplex}(|\mathcal{V}|^{M_Y})} \sum_{y,\mathbf{x} \in D} \ell(\mu; y, \mathbf{x}). \quad (2)$$

We consider the loss to be the negative log-likelihood $\ell(\mu; y, \mathbf{x}) := -\log \mu(y \mid \mathbf{x})$ in this work, and the objective is to minimize the excess risk denoted by $R_{P*}(\mu) - R_{P*}(\mu_*)$.

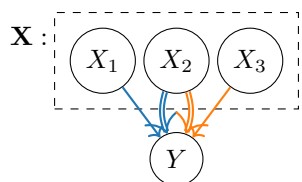

Figure 1: Causal diagrams corresponding to Example 2.1. Color-coded edges show parents of $Y$ in each domain: blue for $\mathcal{M}^1$, orange for $\mathcal{M}^*$. Single edges are the first parent and double edges are the second parent.

We have access to large source data drawn from a set of source domains $\mathcal{M}^1, \mathcal{M}^2, ..., \mathcal{M}^K$ that entail the source distributions $P^1(\mathbf{x}, y), P^2(\mathbf{x}, y), \ldots, P^K(\mathbf{x}, y)$, and our predictions are to be evaluated in the target domain $\mathcal{M}^*$ that entails $P^*(\mathbf{x}, y)$. We assume strictly positive mass for every combination of the variables, i.e., $P^j(\mathbf{x}, y) > \epsilon$ for all $j \in [K] \cup \{*\}$. The data is denoted by $D^1, D^2, \ldots, D^K, D^*$ from the corresponding source and target distributions, where $|D^*| = n$ and $|D^j| = N$ for $j \in [K]$, and typically $N \gg n$. Below is a simple instance of the problem.

**Example 2.1** (Motivating example). Suppose $X_1, X_2, X_3, Y \in \{0, 1, ..., 9\}$. There is a single source domain $\mathcal{M}^1$ and a target domain $\mathcal{M}^*$, described as follows:

$$U_{X_1}, \ldots, U_{X_M} \sim P(u_{X_1}, \ldots, u_{X_m})$$
$$U_Y \sim \text{Multinomial}(\text{prob} : \{0.91, 0.01, ..., 0.01\})$$
$$X_m \leftarrow U_{X_m}, \quad \forall m \in [M]$$
$$Y \leftarrow \begin{cases} X_1 - X_2 + U_Y \pmod{10} & \text{in } \mathcal{M}^1 \\ X_3 - X_2 + U_Y \pmod{10} & \text{in } \mathcal{M}^* \end{cases}$$

The causal diagram corresponding to these SCMs is shown in Figure 1. The causal parents of $Y$ are a different subset of covariates in the source and target domains, but the mechanism that decides $Y$ based on these parents is shared between $\mathcal{M}^*$ and $\mathcal{M}^1$; it is a noisy subtraction of second parent from the first parent.

Suppose we know the parents of $Y$ in both source and target, i.e., we have access to the ordered sets $\mathbf{Pa}_Y^1 = \langle \mathbf{Pa}_Y^1[1], \mathbf{Pa}_Y^1[2] \rangle = \langle X_1, X_2 \rangle$ and $\mathbf{Pa}_Y^* = \langle \mathbf{Pa}_Y^*[1], \mathbf{Pa}_Y^*[2] \rangle = \langle X_3, X_2 \rangle$. Moreover, suppose we know that the *causal module* that generates $Y$ is shared between the two domains, i.e., $f_Y^*(a, b, u_Y) = f_Y^1(a, b, u_Y)$ for all $a, b \in \{0, 1, ..., 9\}$, and $P^*(u_Y) = P^1(u_Y)$. Using this information, we can train a *modular predictor* for $Y$ using data from $\mathcal{M}^1$, i.e., $\mu_1(y \mid a, b) = \hat{P}(y \mid X_1 = a, X_2 = b; D^1)$, and then, because we know the parents of $Y$ in the target, we can plug them into this predictor in the appropriate order and predict $Y$ in the target with small error; this is due to accurate module estimation from the source data. In fact, with large source data, no target data is needed (i.e., zero-shot generalization) once we have such qualitative knowledge about causal structure shared between the domains.

Note that in this case $P^1(Y \mid \mathbf{S}) \neq P^*(Y \mid \mathbf{S})$ for any subset $\mathbf{S} \subset \{X_1, X_2, X_3\}$, i.e., no distributional invariance holds. However, once we unfold $\mathbf{X}$ into $\{X_1, X_2, X_3\}$, the elaborate structure of the parents of $Y$ in each domain and the mechanism sharing enables transport. □

To encode mechanism sharing between the domains, we use the following notation by Correa and Bareinboim (2019); Bareinboim and Pearl (2014).

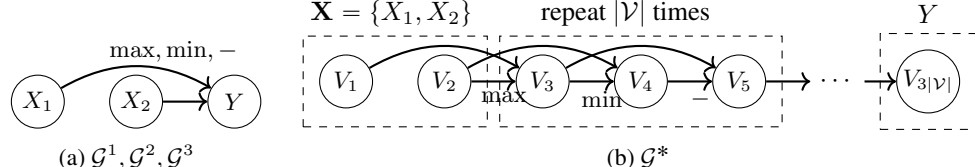

(a) $\mathcal{G}^1, \mathcal{G}^2, \mathcal{G}^3$          (b) $\mathcal{G}^*$

Figure 2: Causal graphs corresponding to Example 2.4. In the sources, the mechanisms determining $Y$ are three noisy operators. The target SCM implements a GCD algorithm via the same three operators through a repeated structure. The query of interest is $P^*(v_{3|\mathcal{V}|} \mid v_1, v_2)$

**Definition 2.2** (Domain discrepancy sets). The collection of subsets of observable variables $\Delta = \{\Delta_{j,j'}\}_{j,j'\in[K]\cup\{*\}}^K$ where $\Delta_{j,j'}$ contains a variable $V \in \mathbf{V}$ if there is a possible mismatch between the causal mechanism of $V$ in domains $\mathcal{M}^j, \mathcal{M}^{j'}$, i.e., either $f_V^j \neq f_V^{j'}$ or $P^j(\mathbf{u}_V) \neq P^{j'}(\mathbf{u}_V)$. □

Notably, because the parents of $Y$ are different between the domains, the existing notions of transportability (e.g., Correa and Bareinboim (2019); Lee et al. (2020)) do not license transport in Example 2.1.

Module-TR (Algorithm 1) is the procedure that generalizes the approach in Example 2.1. First, we identify the source domains $\mathcal{J}$ that share the $Y$ mechanism with the target domain. Next, we pool the data from these relevant sources and reordering the parents to ensure a matching scope for training the predictive module.

---

**Algorithm 1** Module-TR

**Require:** $D^1, D^2, \ldots, D^K, D^*; \Delta, \{\mathbf{Pa}_Y^j\}$
**Ensure:** $\mu_{\text{TR}}(y \mid \mathbf{x}) \approx P^*(y \mid \mathbf{x})$
1: $\mathcal{J} \leftarrow \{j \in [K] \text{ and } Y \notin \Delta_{j,*}\}$
2: $c \leftarrow |\mathbf{Pa}_Y^*|$
3: $D_{\mathbf{R}}^{\text{TR}} \leftarrow \bigcup_{j\in\mathcal{J}} D^j[Y, \mathbf{R}_{1:c} = \mathbf{pa}_Y^j[1:c]]$
4: $D_{\mathbf{R}}^* \leftarrow D^*[Y, \mathbf{R}_{1:c} = \mathbf{Pa}_Y^*[1:c]]$
5: $\mu(y \mid \mathbf{r}) \leftarrow \hat{P}(y \mid \mathbf{r}_{1:c}; D_{\mathbf{R}}^{\text{TR}} \cup D_{\mathbf{R}}^*)$
6: **Return** $\mu_{\text{TR}} \leftarrow \mu(y \mid \mathbf{r} = \mathbf{pa}_Y^*)$

---

**Proposition 2.3** (Module-TR). *In Algorithm 1 with high probability,*

$$R_{P*}(\mu_{\text{TR}}) - R_{P*}(\mu_*) = \begin{cases} \mathcal{O}(\frac{|\mathcal{X}|^c \cdot |\mathcal{Y}|}{\epsilon^2 \cdot N}) & \text{if } \mathcal{J} \neq \varnothing \\ \mathcal{O}(\frac{|\mathcal{X}|^c \cdot |\mathcal{Y}|}{\epsilon \cdot n}) & \text{otherwise} \end{cases} \tag{3}$$

*where $c = |\mathbf{Pa}_Y^*| \leqslant M$.* □

All proofs are in Appendix C. In words, if $\mathcal{J}$ is empty, it means that $Y \in \Delta_{j,*}$ for all source domains $\mathcal{M}^j$, thus no source data can help learning in the target, and in fact, the mechanism of $Y$ in the target domain can be *any* function. Thus, the best error rate decays with $n$ that corresponds to ERM on the target $D^j$. On the other hand, if there are any sources where the mechanism of $Y$ matches with the target, then knowledge of the domain discrepancies and the parent sets allows us to pool the data from relevant sources, reorder the covariates to match with the parents of $Y$ in the target, and train a single predictor using large data of size at least $N$ pooled from the sources and target. In this case, what allows zero-shot generalization is large source data supplemented with domain knowledge about the causal structure within each of the domains and the mechanism discrepancies across the domains.

**Example 2.4** (Beyond module-transportability). Consider three source domain represented by the SCM $\mathcal{M}^1, \mathcal{M}^2, \mathcal{M}^2$, and the target domain represented by $\mathcal{M}^*$, all defined over $X_1, X_2, Y$. The variables have a support of $\{0, 1, \ldots, C\}$, $X_1, X_2$ are independent in all domains and follow a uniform distribution. Suppose the ground truth mechanisms are noisy operators below:

$$f_Y^1(x_1, x_2, u) \approx \max(x_1, x_2), \qquad f_Y^2(x_1, x_2, u) \approx \min(x_1, x_2),$$
$$f_Y^3(x_1, x_2, u) \approx x_1 - x_2, \qquad f_Y^*(x_1, x_2, u) \approx \text{GCD}(x_1, x_2)$$

Here, GCD denotes the greatest common divisor operator. We require them to be noisy to avoid violating the positivity assumption (cf. Preliminaries). Notice that given the parents of $Y$ in both domains and the domain discrepancies, module-TR (Algorithm 1) shall not transport $P^*(y \mid x_1, x_2)$ from the source data, as the function $f_Y^*$ is shared with neither of the source domains.

Interestingly, it is possible to construct a noisy GCD operator by *composition* of the modules that are learnable from the source domains. In particular, one can think of an alternative SCM

$\tilde{\mathcal{M}}^*$ over variables $V_1, V_2, \ldots, V_{3|\mathcal{V}|}$ specified as follows: $V_1, V_2 \sim \text{unif}(\{0, 1, \ldots, |\mathcal{V}|\})$, and for $i \in \{1, 2, \ldots, |\mathcal{V}|\}$

$$V_{3i} \leftarrow \max(V_{3i-1}, V_{3i-2}), \quad V_{3i+1} \leftarrow \min(V_{3i-1}, V_{3i-2}), \quad V_{3i+2} \leftarrow V_{3i} - V_{3i+1}. \quad (4)$$

This SCM in fact corresponds to an algorithm that computes GCD using the $\min, \max, -$ operators, i.e.,

$$P^{\tilde{\mathcal{M}}^*}(v_{3|\mathcal{V}|} \mid v_1, v_2) \approx 1[v_{3|\mathcal{V}|} = \text{GCD}(v_1, v_2)]. \quad (5)$$

If our domain knowledge was so rich to encode this compositional structure and the shared causal mechanisms with the source domains, we could learn the base operators and compose them in the appropriate order to obtain the noisy GCD function. In fact, the query $P^*(y \mid x_1, x_2)$ above is transportable from source data in a broad sense, only with a richer domain knowledge. □

Example 2.4 motivates us to take a more general approach for transportability strategies involving such compositional structures.

Let $\mathbf{V} = \{V_1, V_2, ..., V_{T_j}\}$ denotes the observable variables in SCM $\mathcal{M}^j$, for $j \in [K] \cup \{*\}$, each having a possibly different number of variables. Every $V_i$ takes value in a finite set $\mathcal{V}$, shared across all positions and domains. We assume a causal order $V_1 \prec V_2 \prec ... \prec V_{T_j}$, and that there exists no unobserved confounding. Let $\mathcal{G}^j = \{\mathbf{Pa}_i^j\}_{i=1}^T$ be the causal diagram induced by the SCM $\mathcal{M}^j$ (e.g., Figure 2). The label is $Y : V_{T_*}$ and the covariates are $\mathbf{X} : \{V_1, V_2, \ldots, V_M\}$ in the target domain domain $\mathcal{M}^*$. The goal is still learning $P^*(y \mid \mathbf{x}) = P^*(v_{T_*} \mid \mathbf{v}_{1:M})$ using $N$ labeled data from each of the source domains and $n$ target data. For the especial case of $T_* = M + 1$, this coincides with the module transportability task.

As seen in Example 2.4, the domain discrepancies structure may be more complicated in the sequential setting, allowing a match between mechanisms from different positions $i, i'$ and across different domains $j, j'$. Below is an extension of domain discrepancies useful to accommodate such commonalities.

**Definition 2.5** (Discrepancy oracle). Let $\Delta(i, j; i', j')$ be a boolean function that returns one if either $f_i^j \neq f_{i'}^{j'}$ or $P^j(u_i) \neq P^{j'}(u_{i'})$, and returns zero otherwise. □

Now we can formally define Circuit transportability.

**Definition 2.6.** The query $P^*(v_{T_*} \mid \mathbf{v}_{1:M})$ is circuit-transportable from the source distributions $P^1(\mathbf{v}), \ldots, P^K(\mathbf{v})$ given the discrepancy oracle $\Delta$ and the causal diagrams $\mathcal{G}^1, \ldots, \mathcal{G}^K, \mathcal{G}^*$, if for every tuple of source SCMs $\mathcal{M}^1, \ldots, \mathcal{M}^K$ and target SCM $\mathcal{M}^*$ that induce the causal diagrams and $\Delta$, and also entail the source distributions, we have a unique distribution $P^{\mathcal{M}^*}(v_{T_*} \mid \mathbf{v}_{1:M})$.

In the context of Example 2.4, once we rename $X_1, X_2, Y$ to $V_1, V_2, V_3$ in the source domains, we have $\Delta(3, 3; 5, *) = \Delta(3, 1; 3|\mathcal{V}|, *) = 0$ and $\Delta(3, 2; 6, *) = 1$.

Circuit-TR (Algorithm 2) is a general procedure for circuit-transportability task. The structure encoded by the discrepancy oracle $\Delta$ and the domain-specific causal diagrams $\mathcal{G}^1, \ldots, \mathcal{G}^K, \mathcal{G}^*$ are used to pool data that is causally relevant to each of

---

**Algorithm 2** Circuit-TR

**Require:** $D^1, D^2, \ldots, D^K, D^*; \Delta; \{\mathcal{G}^j : \{\mathbf{Pa}_i^j\}\}$
**Ensure:** $\mu_{\text{TR}}(v_{T_*} \mid v_{1:M}) \approx P^*(v_{T_*} \mid v_{1:M})$
1: **for** $i \in \{M + 1, ..., T_*\}$ **do**
2: $\quad \mathcal{J}_i \leftarrow \{(i', j') : \Delta(i, *; i', j') = 0 \text{ and } j \in [K]\}$
3: $\quad D_i^{\text{TR}} \leftarrow \bigcup_{i', j' \in \mathcal{J}_i \cup \{(i, *)\}} D^{j'}[Y : V_{i'}, \mathbf{X}_{1:c} : \mathbf{Pa}_{i'}^{j'}]$
4: **end for**
5: $\mu_{\text{TR}} = \prod_{i=M+1}^{T_*} \hat{P}(Y = v_i \mid \mathbf{X} = \mathbf{pa}_i^*; D_i^{\text{TR}})$
6: **Return** $\mu_{\text{TR}}(v_{T_*} \mid v_{1:M}) \leftarrow \sum_{v_{M+1:T_*-1}} \mu_{\text{TR}}$

---

the variables in the target domain, aking to module-TR procedure. Next, this pooled data is used to learn the set of conditional distribution $P^*(v_i \mid \mathbf{pa}_i^*)$. These conditional distributions are then composed to yield an estimator of $P^*(v_{M+1:T_*} \mid v_{1:M})$, and finally the variables $V_{M+1}, ..., V_{T_*-1}$ are marginalized out to obtain an estimation of the target quantity $P^*(v_{T_*} \mid v_{1:M})$.

Remark that in module transportability, Algorithm 1 either achieves zero-shot generalization (i.e., rates in terms of $N$) or uses only the target data (rates in terms of $n$) (Lemma B.3 and Proposition 2.3). However, in transporting circuits, the predictor $\mu_{\text{TR}}$ returned by Algorithm 2 may lie in between the two extremes; in particular, if all conditional distributions $\{P^*(v_i \mid \mathbf{Pa}_i^*)\}_{i=M+1}^{T_*}$ are transported,

i.e., data from at least one of the sources is pooled for estimation of each module, then all of them would have low error, resulting in a target risk guarantee for $\mu_{\text{TR}}$ that depends on $N$. On the other hand, if none of the modules $\{P^*(v_i \mid \mathbf{Pa}_i^*)\}_{i=M+1}^{T_*}$ can be transported, then the guarantee would be in terms of the target data size $n$ only. One can imagine in-between situations where some of the conditionals are transported and others must be learned with target data only; in these situations, the risk bound would have a fast and slow components which decay with $n$ and $N$, respectively. Below is an upper-bound for the target risk of structure-informed DA in sequential prediction.

**Theorem 2.7** (Circuit-TR error rate). *In Algorithm 2 with high probability,*

$$
R_{P*}(\mu_{\text{TR}}) - R_{P*}(\mu_*) = \begin{cases} \mathcal{O}(\frac{|\mathcal{V}|^{T*}}{\epsilon^2 N}) & \textit{if } \forall i \in \{M+1, ..., T_*\} : \mathcal{J}_i \neq \varnothing \\ \mathcal{O}(\frac{|\mathcal{V}|^{M+1}}{\epsilon n}) & \textit{otherwise} \end{cases} \tag{6}
$$

*where $T_*$ is the size of the target SCM.*

In words, the guarantee offered by Theorem 2.7 decays with $N$ (i.e., zero-shot generalization) if all components of the sequence from $M+1$ to $T$ can be transported from one of the sources; for a more refined analysis, cf. Appendix E). A transportable circuit can be interpreted as a *causal/mechanistic interpolation* of the source domains, since each target mechanism $f_i^*$ must be present in at least one position of one of the source domains. On the other hand, when at least one component cannot be transported, then the rate would involve a term which decays with $n$, i.e., a slow adaptation.

Generalization to an unseen target domain necessarily relies on some form of domain knowledge or structural assumptions. A natural challenge is that in most realistic settings, such elaborate structural knowledge is not available, however, it is possible to obtain relatively small amount of labeled target data. Can we leverage an implicit structure to transport predictions from the sources to the target for *few-shot generalizations*? In the next section we investigate this challenge.

## 3 FEW-SHOT LEARNING VIA CIRCUIT TRANSPORTABILITY

Suppose we do not have access to a domain knowledge comprised of a collection of causal graphs and the discrepancy oracle. The next example depicts a situation where we can still make accurate predictions in the target domain.

**Example 3.1.** In the context of Example 2.1, suppose we do not have access to domain discrepancy sets $\Delta$ and the parent sets $\mathbf{Pa}_Y^1, \mathbf{Pa}_Y^*$. To compensate for this lack of knowledge, we take the following approach: we train a collection of predictors that would contain at

---

**Algorithm 3** Circuit-AD

**Require:** $D^1, D^2, \ldots, D^K, D^*; \Delta; \{\mathcal{G}^j\}$
**Ensure:** Target classifier $\hat{\mathbb{E}}_{p*}[Y \mid x]$
1: Let $\mathcal{E} = \{(i,j) : i \in [T_j], j \in [K] \cup \{*\}\}$ and $\mathcal{H} \leftarrow \{\}$
2: $D_{\text{tr}}^*, D_{\text{te}}^* \leftarrow \text{partition}(D^*)$
3: **for** every partition of $\mathcal{E}$ into subsets $\mathcal{S} = \{\mathcal{E}_l\}_l$ **do**
4: $\quad \Delta(i,j;i',j') \leftarrow \begin{cases} 0 \text{ if } \exists \mathcal{E}_l \in \mathcal{S} \text{ s.t. } \in (i,j), (i',j') \in \mathcal{E}_l \\ 1 \text{ otherwise} \end{cases}$
5: $\quad$ **for** every set of graphs $\{G^1, \mathcal{G}^2, ..., \mathcal{G}^K, \mathcal{G}^*\}$ **do**
6: $\quad\quad \mathcal{H} \leftarrow \mathcal{H} \cup \{\text{circuitTR}(D^1, ..., D^K, D_{\text{tr}}^*; \Delta; \{\mathcal{G}^j\})\}$
7: $\quad$ **end for**
8: **end for**
9: **Return** $\mu_{\text{AD}} \leftarrow \arg\min_{\mu \in \mathcal{H}} \ell(\mu; D_{\text{te}}^*)$

---

least one candidate whose risk in the target is as good as what is achievable through module-TR, and then we use held-out target data to find the best performing one among these candidates, hopefully identifying the best-performing one with few target samples. In particular, for every $c \in \{0, 1, 2, 3\}$, we take two ordered set of the covariates of size $c$, and regress $Y$ on these $c$ variables in the specified order, once using target data alone and once using source and target data combined. this results in a total of at most $\sum_{c=0}^{3} (\binom{3}{c} \cdot c!)^2 + \binom{3}{c} \cdot c! = 98$ predictors. Notice that this number does not depend on dimensionality of $X, Y$, and is only a function of the number of variables. Finally, we use held-out target data to choose the best performing one in this pool of predictors. This imposes a fixed excess risk on top of what is achievable through module-TR, and since we know that this example is in fact module-transportable, we can guarantee a small total risk with only a few target samples. □

As seen in the example above, each hypothetical domain knowledge yields an estimator for $P^*(v_{T_*} \mid v_{1:M})$ trained using a combination of the source and target data (Algorithm 2). Since the domain knowledge $\Delta, \{\mathcal{G}^j\}_{j \in [K] \cup \{*\}}$ is a discrete object, by fixing $T_*$ which is the size of the target

SCM, there exists finitely many distinct candidate estimators for $P^*(v_{T_*} \mid v_{1:M})$, considering all possibilities. We can partition the target data into training and validation sets of size $\frac{n}{2}$, and use the training part along with the source data to obtain the set of possible estimators of $P^*(v_{T_*} \mid v_{1:M})$. Finally, we can use the target validation data to pick the best performing estimator from the pool. Circuit-AD (Algorithm 3) summarizes this approach, and what follows is its performance guarantee.

**Theorem 3.2** (Circuit-adaptation rate). *Let $\mu_{\mathrm{TR}}, \mu_{\mathrm{AD}}$ be learned by the circuit-TR (Algorithm 2) and circuit-AD (Algorithms 2 and 3). We have,*

$$R_{P*}(\mu_{\mathrm{AD}}) = \mathcal{O}(R_{P*}(\mu_{\mathrm{TR}}) + \sqrt{\frac{K \cdot T_*^3 \cdot \log T_*}{n}}) \tag{7}$$

*where $K$ is the number of sources, and $T_*$ is the size of the target SCM.*

Notably, circuit-AD would have a guarantee that is only marginally worse than what it achievable through the circuit-TR procedure which leverages an elaborate domain knowledge. An important consideration in employing circuit-AD (Algorithm 5) is the choice of $T_*$, that is the number of modules in the target SCM. The example below emphasizes the subtleties of this matter.

**Example 3.3** (Slow rate for GCD). Recall Example 2.4, where GCD deems circuit-transportable from $\max, \min, -$, given, e.g., the causal graph in Figure 2b. Now suppose we do not have access to the causal graphs and domain discrepancies, so we need to use Circuit-AD. We need to make a choice of $T_*$, and circuit-AD offers no basis for this. In hindsight, we can ensure that $T_* = 3|\mathcal{V}|$ is large enough, since there exists a recipe of this length (i.e., causal graph and discrepancies oracle) that licenses circuit-transportability. Due to Theorem 3.2, this choice offers the following guarantee:

$$R_{P*}(\mu_{\mathrm{AD}}) = \mathcal{O}(R_{P*}(\mu_{\mathrm{TR}}) + \sqrt{\frac{|\mathcal{V}|^3 \cdot \log |\mathcal{V}|}{n}}). \tag{8}$$

This is indeed an slow rate, since $n = \Omega(|\mathcal{V}|^3)$ is needed for a constant excess risk; in fact, the regular ERM on target yields similar error rate. □

*How many modules should the target domain have for $P^*(y \mid \mathbf{x})$ to be circuit-transportable?*

This is known as *Minimum Circuit Size Problem (MCSP)* (Shannon, 1949; Lupanov, 1958; Furst et al., 1981; Arora and Barak, 2009; Murray and Williams, 2017; Allender et al., 2018). The modules considered in this circuit complexity theory are often deterministic boolean functions, but, there are also probabilistic extensions too, cf. Adleman (1978); Yao (1977); Håstad (1986). *Clone membership problem (CMP)* (Post, 1941; Lau, 2006; Vollmer, 2009) concerns whether it is even possible to transport a circuit from a fixed set of modules. CMP is indeed *decidable* in time and space polynomial in size of the conditional distribution table (exponential in vocabulary size $|\mathcal{V}|$). In fact, if the source domains offer a functionally complete basis, then its clone contains all functions, meaning that every conditional distribution is circuit transportable with *some* circuit size, but still, adaptation rates depends on the size of the circuit and simply being complete doesn't offer advantage in adaptation. Below is an example that illustrates how a circuit can be adaptable with different error rates from difference source domains.

**Example 3.4** (Fast rate for GCD). In the context of Example 2.4, suppose we also have access to source $\mathcal{M}^4$ where, $f_Y^4 \approx a \bmod b$. An algorithm/circuit for GCD using $\max, \min, \bmod$ is identical to the one constructed with $\max, \min, -$, with just replacing $-$ operator with $\bmod$. This change reduces the number of modules needed to $3\lceil \log_2(|\mathcal{V}|) \rceil$, thus, circuit-AD for GCD using the new source domain has a much better guaranteed excess risk compared to Equation (8), though the guarantee still depends on the vocabulary size. □

Motivated by this observation, we have the following statement.

**Corollary 3.5** (Fast/slow). *Suppose $L$ is the the minimum circuit size for computing $P^*(y \mid \mathbf{x})$ from the basis of modules. if $L$ is constant, then it is possible to achieve very fast adaptation rates. Also, $L = \mathcal{O}(\sqrt[3]{\frac{n}{k}})$ is the threshold of fast/slow adaptability with a fix target dataset of size $n$.*

# 4 ARCHITECTURE, OPTIMIZATION SCHEME, AND SIMULATIONS

Circuit-AD (Algorithm 3) iterates over all combinations of discrepancy oracle and graphs, and this makes it computationally intractable. Here, we introduce a gradient-based alternative.

## 4.1 THE ARCHITECTURE AND PRETRAINING

For simplicity, suppose the same number of variables are observed in all domains, i.e., $T_j = T$. Also, here we discuss the case of $|\mathbf{Pa}_i^j| \leqslant 1$ for all $i \in [T], j \in [K] \cup \{*\}$, though we remove this condition in Appendix G. The goal of pretraining is to use the large source data and learn a set of mappings that satisfy desired properties:

1. The mechanism indicator $\phi : [T] \times [K] \to [d]$, such that,

$$\Phi(i, j) \neq \Phi(i', j') \text{ if and only if } \Delta(i, j; i', j') = 1 \tag{9}$$

2. The parent matrix: $A^j \in [0,1]^{T \times T}$ for all $j \in T$ is lower-diagonal, such that,

$$A_{i,i'}^j = 1 \text{ iff } \mathbf{Pa}_i^j = V_{i'} \tag{10}$$

3. Universal predictor $\Psi : \mathcal{V} \times \mathcal{V} \times [d] \to [0,1]$:

$$\forall y, x \in \mathcal{V}, i \in [T], j \in [K] : \Psi(y \mid x; \phi = \Phi(i,j)) = P^j(V_i = y \mid \mathbf{Pa}_i^j = x) \tag{11}$$

In words, it is desirable that $\Phi(i, j)$ encodes the discrepancy oracle by clustering the position-domain pair into the categories $\{1, ..., d\}$, and $\{A^j\}_{j=1}^K$ encode the causal diagram in each domain. It is clear that once the mappings satisfy the properties Equations (9) to (11), we can have optimal prediction in the source domains: for predicting $v_i$ in domain $\mathcal{M}^j$ take, $\hat{v}_i \sim \Psi(v_i \mid X = A_{i,\cdot}^j \cdot v_{1:T}; \Phi(i,j))$. By design, any instantiation of $\Phi, \{A^j\}, \Psi$ that maximizes a penalized likelihood on source populations satisfies the above properties, as stated below.

**Proposition 4.1** (Pretraining). *Let $\theta^{\mathrm{src}}$ be a set of parameters for the mappings above such that,*

$$\theta^{\mathrm{src}} \in \arg\min_{\theta \in \Theta} \Big( \sum_{|\mathcal{V}|^T} \sum_{j=1}^K P^j(v_t) \sum_{i=1}^T -\log \Psi_\theta(v_i \mid A_{\theta\,i,\cdot}^j \cdot v_{1:T}; \Phi_\theta(i,j)) \Big) + \lambda(d + \|A^\cdot\|_1), \tag{12}$$

*where $\Theta$ denotes all parameterizations for the mappings, $[d]$ is the range of $\Phi$, and $\|A^\cdot\|_1$ denotes sum of the entries of the parent matrices. $\Phi_{\theta^{\mathrm{src}}}, \{A_{\theta^{\mathrm{src}}}^j\}, \Psi_{\theta^{\mathrm{src}}}$ satisfy Equations (9) to (11).* □

We experiment with a synthetic example with $T_1, T_* = 10$, and validate that the parent matrices are indeed learned during the pretraining, cf. Figure 3. Notably, Nichani et al. (2024) illustrates a similar phenomenon, where the transformer architecture learns such sequential dependencies. Here, we show that this custom architecture captures not only the causal dependencies, but also the source-source discrepancy oracle.

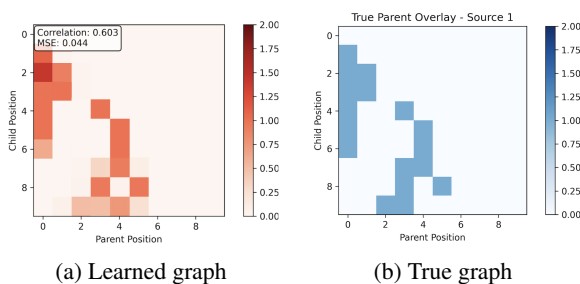

(a) Learned graph          (b) True graph

Figure 3: Implicit causal discovery in pretraining.

## 4.2 FINE-TUNING

We partition the target data $D^*$ into $D_{\mathrm{tr}}^*, D_{\mathrm{ft}}^*, D_{\mathrm{te}}^*$ of proportionate size. Recall $[d]$ as the range of the mapping $\Phi$. Once pretrained, each $\phi \in [d]$ corresponds to a subset of position-domain pairs in the source that share the causal mechanism; this is due to Proposition 4.1 that ensures Equation (9). Suppose for a position $i \in [T]$, it holds that $\Delta(i, *; i', j') = 0$. Thus, for all $x, y \in \mathcal{V}$,

$$P^*(V_i = y \mid \mathbf{Pa}_i^* = x) = P^{j'}(V_{i'} = y \mid \mathbf{Pa}_i^{j'} = x) \quad (\Delta(i, *; i', j') = 0) \tag{13}$$

$$= \Psi_{\theta^{\mathrm{src}}}(V_{i'} = y \mid X = x; \phi = \Phi(i', j')) \quad (\text{Equation (11)}) \tag{14}$$

Notably, $\Psi_{\theta^{\mathrm{src}}}$ is learned in the pretraining stage, yet we need to discover $\phi, \mathbf{Pa}_i^*$ at each position $i \in [T]$. To this end, we take a target parent matrix $A^* \in [0,1]^{T \times T}$ to encode $\mathbf{Pa}_i^*$ via $A_{i,i'}^* = 1$ if $\mathbf{Pa}_i^* = V_{i'}$, and also a target mechanism indicator $\Phi^* : [T] \to [d]$. Next, we use $D_{\mathrm{ft}}^*$ to learn,

$$\theta^{\mathrm{trg}} \in \arg\min_{\theta \in \Theta} \sum_{v_{1:T} \in D_{\mathrm{ft}}^*} \sum_{i=1}^T -\log \Psi_{\theta^{\mathrm{src}}}(Y = v_i \mid X = A_{\theta\,i,\cdot}^* \cdot v_{1:T}; \Phi_\theta^*(i)). \tag{15}$$

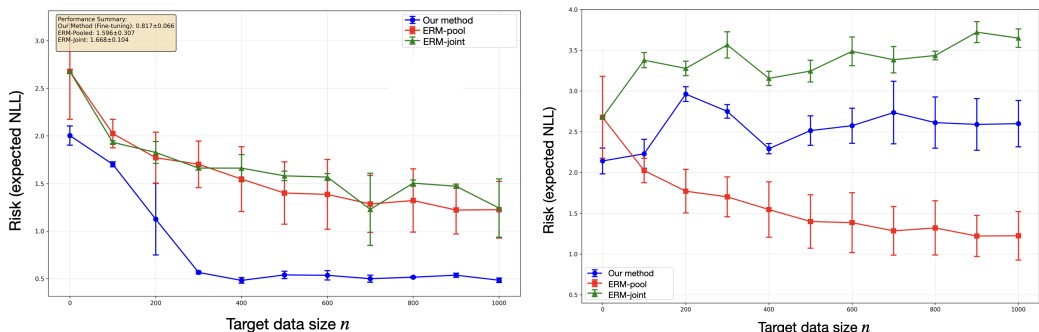

(a) Adapting fast due to process supervision  (b) Adapting slow due to no process supervision

Figure 4: The performance of our method which is based on structure agnostic domain adaptation, in comparison with the baselines that train jointly on source and target data, either by discarding the domain indices (ERM-pool) or by keeping them (ERM-joint).

This optimization considered at each position $i \in [T]$ is equivalent to using $|D_{\mathrm{ft}}^*|$ target data to pick the best-performing predictor for $V_i$ in $\mathcal{M}^*$ among a pool of $d \cdot (i-1)$ candidates from the sources.

We use $D_{\mathrm{tr}}^*$ to learn a separate target-only model at every position $i$; in particular, let $\mu_i^* := \hat{P}(v_i \mid v_{1:i-1}; D_{\mathrm{tr}}^*)$. Finally, at each position $i$, we choose a linear interpolation of the best transported predictor and $\mu_i^*$. This is performed via learning the *transport indicators* $s_1^*, \ldots, s_T^* \in [0,1]$ through,

$$s_i^* \in \underset{s \in [0,1]^T}{\arg\min} \sum_{v \in D_{\mathrm{te}}^*} \sum_{i=1}^{T} -\log(s_i \cdot \Psi_{\theta^{\mathrm{src}}}(v_i \mid A_{\theta^{\mathrm{trg}} i,\cdot}^* \cdot v_{1:T}; \Phi_{\theta^{\mathrm{trg}}}^*(i)) + (1-s_1) \cdot \mu_i^*(v_i \mid v_{1:i-1})) \quad (16)$$

In words, $s_i \approx 0$ indicates that the target-only model performs best on the held-out data $D_{\mathrm{te}}^*$, thus we decide on no transport from the sources. On the other hand, $s_i \approx 1$ indicates a decision to transport. Finally, we compute the estimation of the query of interest $P^*(v_t \mid v_{1:M})$:

$$\hat{\mu}_{\mathrm{ft}}(v_T \mid v_{1:M}) = \sum_{v_{M+1:T-1}} \prod_{i=M+1}^{T} s_i \cdot \Psi_{\theta^{\mathrm{src}}}(v_i \mid A_{\theta^{\mathrm{trg}} i,\cdot}^* \cdot v_{1:T}; \Phi_{\theta^{\mathrm{trg}}}^*(i)) + (1-s_i) \cdot \mu_i^*(v_i \mid v_{1:i-1})$$

What follows justifies equivalence of two-stage adaptation with Algorithm 3.

**Proposition 4.2** (Fine-tuning rate). *Let $\mu_{\mathrm{AD}}$ be learned by Algorithm 3 and $\mu_{\mathrm{ft}}$ (Section 4.2) be the result of the two-stage adaptation. We have, $R_{P*}(\mu_{\mathrm{ft}}) = \mathcal{O}(R_{P*}(\mu_{\mathrm{AD}}))$*

We consider our method in comparison with two baselines:

1. **ERM-pool.** We drop the domain indices and pool data from all domains.

2. **ERM-joint.** We keep the domain indices, and treat the target as another source in pretraining.

We use the same architecture for both baselines to isolate adaptability. The goal is learning $P^*(v_{10} \mid v_{1:5})$, and we deliberately picked a circuit-transportable instance; see the graphs in Appendix F. Witnessed by Figure 4a, circuit-AD surpasses the baselines when the circuit size matches with a ground truth (i.e., $T_* = 10$), however, for a restrictive choice of $T_* = 6$ (i.e., module-TR), the task is not transportable anymore, thus, circuit-AD achieves a poor performance, as predicted by the theory.

## 5 CONCLUSIONS

We proposed a causal framework for learning from different domains. Circuit-TR is an extension to the causal transportability theory for compositional generalization task. We devise circuit-AD, a supervised domain adaptation scheme that uses circuit-TR as a subroutine to mitigate challenges of missing domain knowledge. Our findings draw a connection between minimum circuit size problem and error rates associated with circuit-AD algorithm. To address intractability of the symbolic algorithms, we introduced a transformer-like architecture and training agenda that mimics an exhaustive search procedure of circuit-AD, and show its validity with controlled experiments.

## ETHICS & REPRODUCIBILITY STATEMENTS

We confirm that this research adheres to all requirements outlined in the ICLR Code of Ethics. The study does not include human participants, personally identifiable information, or sensitive use cases, and we do not anticipate any direct ethical concerns arising from this work.

All assumptions are clearly stated in the paper, and all theoretical results have complete proofs, presented Appendix C. Detailed description of the code implementation is presented in Appendix F.1 to ensure Reproducibility.

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

# Supplementary Material

## A  LLM USAGE DISCLOSURE

In this work, we used Chat-gpt and Claude, and Cursor for the following purposes:

- Ideation, choice of notation, and structuring the paper.

- Literature overview and references.

- Developement of the codebase, and illustration/visualization.

- Content on reproducibility and implementation details in Appendix F.1.

## B  SIMPLIFIED UNI-CAUSE TRANSPORTABILITY AND ADAPTATION

Consider the classification task where $Y$ takes value in a finite support $\mathcal{Y}$, and the covariate $X$ that takes value in a finite support $\mathcal{X}$; in particular the objective is learning $P^*(Y = y \mid X = x)$ within the hypothesis class containing all functions $\mu(y \mid x) : \mathcal{X} \to \text{simplex}^{|\mathcal{Y}|}$. There is a loss function $\ell(\mu; y, \mathbf{x})$, and the risk is defined as the expected loss $R_{P*}(\mu) := \mathbb{E}_{P*}[\ell(\mu; Y, \mathbf{X})]$. The true risk minimizer is denoted as $\mu_* \in \arg\min_{\mu:\mathcal{X}\to\text{simplex}^{|\mathcal{Y}|}} R_{P*}(\mu)$, and the empirical risk minimizer w.r.t. data $D$ is denoted as,

---

**Algorithm 4** simple-AD $(X \to Y)$

---

**Require:** $D^1, D^2, \ldots, D^K, D^*; \Delta$
**Ensure:** $\mu_{\text{TR}} \approx P^*(y \mid x)$
1: $\mathcal{J} \leftarrow \{j \in [K] \text{ and } Y \notin \Delta_{j,*}\}$
2: $D^{\text{TR}} \leftarrow \bigcup_{j\in[K] \text{ s.t. } Y\notin\Delta_{j,*}} D^j$
3: **Return** $\mu_{\text{TR}} \leftarrow \hat{P}(y \mid x; D^{\text{TR}} \cup D^*)$

---

$$\hat{P}(y \mid \mathbf{x}; D) \in \underset{\mu:\mathcal{X}\to\text{simplex}^{|\mathcal{Y}|}}{\arg\min} \sum_{y,\mathbf{x}\in D} \ell(\mu; y, \mathbf{x}). \tag{17}$$

We consider the loss to be the negative log-likelihood $\ell(\mu; y, x) := -\log\mu(y \mid \mathbf{x})$ in this work, and the objective is to minimize the excess risk denoted by $R_{P*}(\mu) - R_{P*}(\mu_*)$.

Suppose we have access to target data $D^*$ drawn i.i.d. from the target domain $\pi^*$ that entails the target distribution $P^*(x, y)$, as well as source data $D^1, D^2, \ldots, D^K$ from a set of source domains $\Pi^{\text{src}} = \{\pi^1, \pi^2, \ldots, \pi^K\}$ that entail the source distributions $P^{\text{src}} = \{P^1(x, y), P^2(x, y), \ldots, P^K(x, y)\}$. Let $n = |D^*|$ and $N = |D^j|$ for all $j \in [K]$, and suppose $N \gg n$. We assume strictly positive mass for every combination of the variables, i.e., $P^j(x, y) > \epsilon$ for all $j \in [K] \cup \{*\}$.

**Example B.1** (Classification in $X \to Y$ case). Suppose the source domains are governed by SCMs $\mathcal{M}^1, \mathcal{M}^2, \ldots, \mathcal{M}^K$, and the target domain $\pi^*$ is governed by the SCM $\mathcal{M}^*$; For domain $\pi^j$ the SCM $\mathcal{M}^j$ is denoted as follow:

$$U_X, U_Y \sim \text{unif}([0, 1])$$
$$X \leftarrow f_X^j(U_X)$$
$$Y \leftarrow f_Y^j(X, U_Y).$$

The source and target SCMs all induce the same causal diagram $X \to Y$, which indicates that $X$ is the cause of $Y$, and no unobserved confounders are present. Notice that without further assumptions the source data is unrelated to classification in the target; for example, in a case of $X \in \{0, 1\}$ it is possible that $f_Y^1 = f_Y^2 = \ldots = f_Y^K : 1_{\{U_Y > 0.5\}}$, which means that $Y \perp\!\!\!\perp X$ across all sources, but $f_Y^*(x, u_Y) = X \oplus 1_{\{U_Y > 0.9\}}$, which means that $P^*(Y = 1 \mid X = 0) = 0.1$ and $P^*(Y = 1 \mid X = 1) = 0.9$ in the target domain. □

In the next example we use the domain discrepancy sets in an instance of the domain adaptation task.

**Example B.2** ($\Delta$ might allow direct-transport, or suggest no transport). In the context of Example B.1, suppose we have access to $\Delta$. If there exists $j \in [K]$ such that $Y \notin \Delta_{j,*}$, then $f_Y^* = f_Y^j$ and

$P^*(u_Y) = P^j(u_Y)$, and therefore,

$$P^*(y \mid x) = \sum_{u_Y \in \mathcal{Y}} P^*(y \mid u_Y, x) \cdot P^*(u_Y \mid x) \qquad \text{(introduce } U_Y)$$

$$= \sum_{u_Y \in \mathcal{Y}} 1_{\{f_Y^*(x, u_Y) = y\}} \cdot P^*(u_Y) \qquad \text{(defn. of } f_Y^* \And U_Y \perp\!\!\!\perp X)$$

$$= \sum_{u_Y \in \mathcal{Y}} 1_{\{f_Y^j(x, u_Y) = y\}} \cdot P^j(u_Y) = P^j(Y \mid x) \qquad (Y \notin \Delta_{j,*})$$

Thus, to predict the label in the target, it suffices to estimate $P^j(Y \mid x)$ using large source data available from $\pi^j$. However, if $Y \in \Delta_{j,*}$ for all $j \in [K]$, then domain discrepancy sets reject use of any source data, as the real SCMs may be similar to what was discussed in Example B.1. □

A simple procedure in Algorithm 4 describes the above approach, and below is a guarantee.

**Lemma B.3** (Simple-TR; $X \to Y$ case). *In Algorithm 4 with high probability,*

$$R_{P*}(\mu_{\mathrm{TR}}) - R_{P*}(\mu^*) = \begin{cases} \mathcal{O}(\frac{|\mathcal{X}| \cdot |\mathcal{Y}|}{\epsilon^2 \cdot N}) & \text{if } \mathcal{J} \neq \varnothing \\ \Omega(\frac{|\mathcal{X}| \cdot |\mathcal{Y}|}{\epsilon \cdot n}) & \text{otherwise} \end{cases} \tag{18}$$

*where $\mathcal{J}$ is obtained through Algorithm 4.* □

In words, if the mechanism of $Y$ in the target matches with that of any of the sources, then because of support overlap (due $P(x,y) \geqslant \epsilon$) simple-TR can achieve fast rates that depend on the source data size $N$ which is typically large. This case is called *transportable* in the causal inference literature, or domain generalization or zero-shot learning in the literature. In the other cases (i.e., the non-transportable scenario), even with access to the structure $\Delta$, there might exist vastly different realities (i.e., a tuple of source and target SCMs) which admit the structural assumption encoded in $\Delta$ but they do not agree on the classification rule in the target domain, thus, no adaptation is possible. These two extreme cases happen as a byproduct of the discrete nature of the domain discrepancy sets; if $Y \notin \Delta_{j,*}$ then the mechanism determining $Y$ matches perfectly between $\pi^*, \pi^j$, and since there is no confounding between $Y$ and $X$ (i.e., $U$ variable pointing to both $X$ and $Y$), we ensure that $P^*(y \mid x) = P(y \mid x)$. Note that not having a confounder is critical to this conclusion based on the structure $\Delta$; in Appendix D, we discuss how confounders can complicate simple-TR.

Notably, the risk upper-bound provided in Lemma B.3 is not tight, e.g., $\epsilon^2$ in the denominator of the transportable case can be improved through adaptive procedures. Since we are assuming that $\epsilon$ is a constant and $N \gg n$, the bounds serve the purpose for this work. Our main focus throughout is identifying which source domains contain useful information for prediction in the target, and how that information can be incorporated in learning, given the structure $\Delta$, thus we rely on the covariates overlap. On the other hand, in many theoretical work on domain adaptation, it is presumed that the sources are all useful for prediction in target, e.g., there exists a unique best hypothesis $h^*$ for all domains Ben-David et al. (2006); Mansour et al. (2009b), thus, in these work the main complexity of DA comes from lack of overlap between the covariate distributions across the domains.

We treat the simple-TR procedures (such as Algorithm 4) as the best one can do given knowledge of the structural properties of the problem. In reality, access to such structure may not be viable, and in the next example we would like to consider adaptation in situations where $\Delta$ is unknown.

**Example B.4** (Agnostic approach; $X \to Y$ case). In the context of Example B.1, suppose we the structure $\Delta$ is unknown, yet

---

**Algorithm 5** simple-AD

**Require:** $D^1, D^2, \ldots, D^K, D^*$
**Ensure:** $\mu_{\mathrm{AD}}(y \mid x) \approx P^*(y \mid x)$
 1: $D_{\mathcal{D}}^*, D_{\mathcal{D}_{\mathrm{test}}}^* \leftarrow \text{partition}(D^*)$
 2: $\psi_{\mathcal{S}} \leftarrow \hat{P}(y \mid x; D_{\mathcal{D}}^* \cup \bigcup_{j \in \mathcal{S}} D^j)$ for all $\mathcal{S} \subseteq [K]$
 3: **Return** $\mu_{\mathrm{AD}} \leftarrow \arg\min_{\mathcal{S} \subseteq [K]} \ell(\psi_{\mathcal{S}}; D_{\mathcal{D}_{\mathrm{test}}}^*)$

---

we would like to achieve guarantees not much worse than what is achievable using $\Delta$. Note that simple-AD (Algorithm 4) pools data from the source domain $\pi^j$ with the target data whenever $\Delta$ implies that $P^*(y \mid x) = P^j(y \mid x)$. If any of the source data is pooled, then the error decays with $N$ which is typically very large, otherwise, it would decay with $n$, the number of target data points.

We can take the following approach to benefit from the source data even without the structure. Partition the target data into $D^* = D_{\mathcal{D}}^* \sqcup D_{\mathcal{D}_{\mathrm{test}}}^*$ of equal size $\frac{n}{2}$. Then, for each subset of the

sources such as $\mathcal{S} \subseteq [K]$, learn a predictor $\psi_{\mathcal{S}}(y \mid x) := \hat{P}(y \mid x; D^*_{\mathcal{D}} \cup \bigcup_{j \in \mathcal{S}} D^j)$. Finally, use $D^*_{\mathcal{D}_{\text{test}}}$ to choose the best of the $2^K$ predictors learned from each of the domains, i.e., $\mu_{\text{AD}} \leftarrow \arg\max_{\psi_{\mathcal{S}}:\mathcal{S} \subseteq [K]} \sum_{y,x \in D^*_{\mathcal{D}_{\text{test}}}} \ell(\psi_{\mathcal{S}}; y, x)$. The best error achievable using the structure $\Delta$ can be achieved by at least one of the $2^K$ predictors we have learned. Thus, the extra risk of the above procedure compared to the simple-TR is equivalent to learning from a finite hypothesis class of size $2^K$ using $D^*_{\mathcal{D}_{\text{test}}}$ data, which is bounded by $\mathcal{O}(\sqrt{\frac{K}{n}})$. □

Example B.4 shows that through a simple training-validation procedure, without access to the domain discrepancies, it is possible to achieves rates that are only slightly worse than what is achievable through explicit access to the domain discrepancies $\Delta$. We call such strategies *Agnostic* throughout this work, and Algorithm 5 summarizes this approach. What follows is a formal statement.

**Theorem B.5** (simple-AD error rate). *Let $\mu_{\text{TR}}, \mu_{\text{AD}}$ be learned using simple-TR and simple-AD (Algorithms 4 and 5), respectively. We have,*

$$R_{P*}(\mu_{\text{AD}}) = \mathcal{O}(R_{P*}(\mu_{\text{TR}}) + \sqrt{\frac{K}{n}}), \tag{19}$$

*where $K$ is the number of source domains, and $n$ is the number of target data.* □

## C  PROOFS

**Definition C.1** (strongly convex functions). *A function $f(x)$ is $m$-strongly convex if,*

$$f(x') \geqslant f(x) + \nabla f(x)^T(x' - x) + \frac{m}{2}\|x' - x\|^2, \quad \forall x, x' \in \mathcal{X}, \tag{20}$$

□

Next, we show that the risk in our problem is strongly convex under strict positivity assumption $P^*(x,y) > \epsilon$.

**Lemma C.2** (strongly convex risk.). *$R_{P*}(\mu)$ is $\epsilon$-strongly convex w.r.t. $\mu$ under the assumption of $P^*(x,y) > \epsilon$.*

*Proof.* Recall the loss function $\ell(\mu; y, x) = -\log \mu(y \mid x)$. Thus, the true risk of $\mu_{\text{TR}}$ can be expressed as:

$$R_{P*}(\mu_{\text{TR}}) = \mathbb{E}_{P*}[-\log \mu_{\text{TR}}(Y \mid X)] \tag{21}$$

$$= \sum_x P^*(x) \cdot \sum_y P^*(y \mid x) \cdot \log \frac{1}{\mu_{\text{TR}}(y \mid x)} \tag{22}$$

$$= \sum_x P^*(x) \cdot D_{D_{\text{KL}}}\big(P^*(\cdot \mid x)\|\mu_{\text{TR}}(\cdot \mid x)\big). \tag{23}$$

For a fixed $x \in \mathcal{X}$, the KL loss $D_{D_{\text{KL}}}\big(P^*(\cdot \mid x)\|\mu_{\text{TR}}(\cdot \mid x)\big)$ is 1-strongly convex for the interior of the simplex, i.e., under positivity. Thus, the weighted sum $\sum_x P^*(x) \cdot D_{D_{\text{KL}}}\big(P^*(\cdot \mid x)\|\mu_{\text{TR}}(\cdot \mid x)\big)$. with $P^*(x) > \epsilon$ would be $\epsilon$-strongly convex Boyd and Vandenberghe (2004). □

What follows is standard high-probability bound for excess risk of ERM with strongly convex risk, adapted from Kakade and Tewari (2009).

**Corollary C.3.** *Let the ERM solution be,*

$$\mu_{\text{ERM}} := \hat{P}(y \mid x; D^*) \in \arg\min_{\mu:\mathcal{X}\to\text{simplex}^{|\mathcal{Y}|}} \sum_{x,y \in D^*} -\log \mu(y \mid x), \tag{24}$$

*and let the true risk minimizer be,*

$$\mu_* = \arg\min_{\mu:\mathcal{X}\to\text{simplex}^{|\mathcal{Y}|}} \mathbb{E}_{Y,X \sim P*}[-\log \mu(y \mid x)]. \tag{25}$$

*Under $P^*(x, y) > \epsilon$, for any $\delta > 0$ the following holds with probability $1 - \delta$:*

$$R_{P*}(\mu_{\mathrm{ERM}}) - R_{P*}(\mu_*) \leqslant \frac{8 \cdot \left( \ln \frac{1}{\delta} + |\mathcal{X}| \cdot |\mathcal{Y}| \cdot \ln(1 + \frac{n}{|\mathcal{X}| \cdot |\mathcal{Y}|}) \right)}{\epsilon \cdot n} = \mathcal{O}(\frac{|\mathcal{X}| \cdot |\mathcal{Y}|}{\epsilon \cdot n}), \tag{26}$$

*where $n = |D^*|$.* □

We also show the following bound for the risk of the transported estimators.

**Lemma C.4.** *Suppose $P^*(y \mid x) = P(y \mid x)$. Define the transported predictor as the ERM over $D \sim P(x, y)$:*

$$\mu_{\mathrm{TR}} := \hat{P}(y \mid x; D) \in \underset{\mu: \mathcal{X} \to \mathrm{simplex}^{|\mathcal{Y}|}}{\arg\min} \sum_{x, y \in D} -\log \mu(y \mid x), \tag{27}$$

*and let the true risk minimizer be defined in Equation (25). Suppose $P^*(x, y), P(x, y) > \epsilon$. For any $\delta > 0$ the following holds with probability $1 - \delta$:*

$$R_{P*}(\mu_{\mathrm{TR}}) - R_{P*}(\mu_*) = \mathcal{O}(\frac{|\mathcal{X}| \cdot |\mathcal{Y}|}{\epsilon^2 \cdot N}), \tag{28}$$

*where $N = |D|$.*

*Proof.* Equal conditionals, i.e., $P^*(y \mid x) = P(y \mid x)$, implies that the true risk minimizer matches under both distributions, i.e.,

$$\mu_* \in \arg\min_\mu \mathbb{E}_{P*}[\ell(\mu; y, x)] \iff \mu_* \in \arg\min_\mu \mathbb{E}_P[\ell(\mu; y, x)]. \tag{29}$$

Since $\mu_{\mathrm{TR}}$ is the solution of ERM under $P$, based on Corollary C.3, we have:

$$R_P(\mu_{\mathrm{TR}}) - R_P(\mu_*) = \mathcal{O}(\frac{|\mathcal{X}| \cdot |\mathcal{Y}|}{\epsilon \cdot N}). \tag{30}$$

Let $\alpha = \max_{x \in \mathcal{X}} \frac{P^*(x)}{P(x)}$. For any $\mu : \mathcal{X} \to \mathrm{simplex}^{|\mathcal{Y}|}$, we related the risk under $P$ and $P^*$:

$$R_{P*}(\mu_{\mathrm{TR}}) - R_{P*}(\mu_*) = \sum_x P^*(x) \cdot \sum_y P(y|x) \cdot (\log \mu_*(y|x) - \log \mu_{\mathrm{TR}}(y|x)) \tag{31}$$

$$= \sum_x \frac{P^*(x)}{P(x)} \cdot P(x) \cdot \sum_y P(y|x) \cdot (\log \mu_*(y|x) - \log \mu_{\mathrm{TR}}(y|x)) \tag{32}$$

$$= \mathbb{E}_{(X,Y) \sim P} \left[ \frac{P^*(X)}{P(X)} \cdot (\log \mu_*(y|x) - \log \mu_{\mathrm{TR}}(y|x)) \right] \tag{33}$$

$$\leqslant \alpha \cdot \mathbb{E}_{(X,Y) \sim P} \left[ \log \mu_*(y|x) - \log \mu_{\mathrm{TR}}(y|x) \right] \tag{34}$$

$$= \alpha \cdot \left( R_P(\mu_{\mathrm{TR}}) - R_P(\mu_*) \right) \tag{35}$$

$$= \mathcal{O}(\frac{\alpha \cdot |\mathcal{X}| \cot |\mathcal{Y}|}{\epsilon \cdot N}) = \mathcal{O}(\frac{|\mathcal{X}| \cot |\mathcal{Y}|}{\epsilon^2 \cdot N}). \tag{36}$$

The last line follows from strict positivity:

$$\alpha = \max_{x \in \mathcal{X}} \frac{P^*(x)}{P(x)} \leqslant \frac{\max_{x \in \mathcal{X}} P^*(x)}{\min_{x \in \mathcal{X}} P(x)} \leqslant \frac{1}{\epsilon} \tag{37}$$

□

## C.1 PROOF OF LEMMA B.3

If $\mathcal{J} = \varnothing$, then the algorithm learns $\mu_{\mathrm{TR}} \leftarrow \hat{P}(y \mid x; D^*)$, i.e., ERM on the target data only. Followed from Corollary C.3, we obtain the desired guarantee.

If $\mathcal{J} \neq \varnothing$, then there exists at least one source domain $j \in \mathcal{J}$ for which $P^j(y \mid x) = P^*(y \mid x)$, and we transport the predictor from that domain. Since $|D^j| = N$, using Lemma C.4, we obtain the desired result.

## C.2 PROOF OF THEOREM B.5

In Algorithm 5 we first compute a collection of predictors $\{\psi_{\mathcal{S}}\}_{\mathcal{S} \subseteq [K]}$. Then we use held-out target data to choose the one with smallest risk. Let,

$$\tilde{\mu} \in \underset{\mu \in \{\psi_{\mathcal{S}}\}_{\mathcal{S} \subseteq [K]}}{\arg\min} \; R_{P*}(\mu), \tag{38}$$

And let $\mu_{\mathrm{TR}}$ be obtained from Algorithm 4. Firstly, consider the following cases:

1. $\mathcal{J} = \varnothing$: In this case, $\mu_{\mathrm{TR}}$ is obtained through ERM on the target data, i.e., $\mu_{\mathrm{TR}} = \hat{P}(y \mid x; D^*)$, achieving the following guarantee (Corollary C.3):

$$R_{P*}(\mu_{\mathrm{TR}}) - R_{P*}(\mu_*) = \mathcal{O}(\frac{|\mathcal{X}| \cdot |\mathcal{Y}|}{\epsilon \cdot n}). \tag{39}$$

Notably, for $\mathcal{S} = \varnothing$, we have $\psi_{\mathcal{S}} = \hat{P}(y \mid x; D_{\mathcal{D}}^*)$, where $|D_{\mathcal{D}}^*| = \frac{n}{2}$. Thus,

$$R_{P*}(\tilde{\mu}) - R_{P*}(\mu_*) \leqslant R_{P*}(\psi_{\varnothing}) - R_{P*}(\mu_*) \tag{40}$$

$$= \mathcal{O}(\frac{|\mathcal{X}| \cdot |\mathcal{Y}|}{\epsilon \cdot n}). \tag{41}$$

2. $\mathcal{J} \neq \varnothing$: In this case, $\mu_{\mathrm{TR}}$ is obtained through ERM on source data from domains $\pi^j$ for $j \in \mathcal{J}$, which would achieving the following guarantee (Lemma C.4):

$$R_{P*}(\mu_{\mathrm{TR}}) - R_{P*}(\mu_*) = \mathcal{O}(\frac{|\mathcal{X}| \cdot |\mathcal{Y}|}{\epsilon^2 \cdot N}). \tag{42}$$

Notably, for $\mathcal{S} = \mathcal{J}$, we have $\psi_{\mathcal{S}}$ trained using the same pooled data as $\mu_{\mathrm{TR}}$, where $|D_{\mathcal{D}}^*| = \frac{n}{2}$. Thus,

$$R_{P*}(\tilde{\mu}) - R_{P*}(\mu_*) \leqslant R_{P*}(\psi_{\mathcal{J}}) - R_{P*}(\mu_*) \tag{43}$$

$$= \mathcal{O}(\frac{|\mathcal{X}| \cdot |\mathcal{Y}|}{\epsilon^2 \cdot N}). \tag{44}$$

Comapring these rates with Lemma B.3 confirms:

$$R_{P*}(\tilde{\mu}) = \mathcal{O}(R_{P*}(\mu_{\mathrm{TR}})). \tag{45}$$

Next, we show that empirical version of $\tilde{\mu}$, namely $\mu_{\mathrm{AD}}$, achieves the desirable excess compared to $\tilde{\mu}$.

$\mu_{\mathrm{AD}}$ in Algorithm 5 is achieved by minimizing the empirical risk over the finite collection $\{\psi_{\mathcal{S}}\}_{\mathcal{S} \subseteq [K]}$ using data $D_{\mathcal{D}_{\mathrm{test}}}^*$ of size $\frac{n}{2}$. Standard uniform convergence guarantees of finite hypothesis classes (Vapnik and Chervonenkis (1971); Shalev-Shwartz and Ben-David (2014)) imply that for any $\delta > 0$, with probability $1 - \delta$ the excess risk can be upper-bounded as:

$$R_{P*}(\mu_{\mathrm{AD}}) - R_{P*}(\tilde{\mu}) \leqslant \sqrt{\frac{\log |\{\psi_{\mathcal{S}}\}_{\mathcal{S} \subseteq [K]}|}{2 \cdot \frac{n}{2}}} + \sqrt{\frac{\log(1/\delta)}{2 \cdot \frac{n}{2}}}. \tag{46}$$

This is due to the fact that $|\{\psi_{\mathcal{S}}\}_{\mathcal{S} \subseteq [K]}| = \mathcal{O}(2^K)$. Finally, Equation (45) implies,

$$R_{P*}(\mu_{\mathrm{AD}}) \leqslant R_{P*}(\tilde{\mu}) + \sqrt{\frac{\log |\{\psi_{\mathcal{S}}\}_{\mathcal{S} \subseteq [K]}|}{2 \cdot \frac{n}{2}}} + \sqrt{\frac{\log(1/\delta)}{2 \cdot \frac{n}{2}}} \tag{47}$$

$$= \mathcal{O}(R_{P*}(\mu_{\mathrm{TR}}) + \sqrt{\frac{K}{n}}) \tag{48}$$

## C.3 PROOF OF PROPOSITION 2.3

Proof follows the logic of the proof of Lemma B.3 (Appendix C.1). In the transportable case we would have $\mathcal{J} \neq \varnothing$ in Algorithm 1. Thus, the data used for estimation of $\mu_{\mathrm{TR}}$ is pooled from at least

one source domain $\pi^j$ for $j \in \mathcal{J}$, where $|D^j| = N$. The conditional distribution to be estimated is $\mu_{\text{TR}}(y \mid \mathbf{pa}_Y^*)$, and for $c = |\mathbf{Pa}_Y^*|$, following Lemma C.4, we get the bound,

$$R_{P*}(\mu_{\text{TR}}) - R_{P*}(\mu_*) = \mathcal{O}(\frac{|\mathcal{X}|^c \cdot |\mathcal{Y}|}{\epsilon^2 \cdot N}). \tag{49}$$

On the other hand, in the non-transportable case, we would have $\mathcal{J} = \varnothing$, thus $\mu_{\text{TR}}$ is trained using only data from the target domain, and due to Corollary C.3, achieves,

$$R_{P*}(\mu_{\text{TR}}) - R_{P*}(\mu_*) = \mathcal{O}(\frac{|\mathcal{X}|^c \cdot |\mathcal{Y}|}{\epsilon \cdot n}). \tag{50}$$

## C.4 Proof of Theorem 2.7

The query of interest $P^*(v_T \mid v_{1:M})$ can be computed through the following formula by introducing the intermediate variables $V_{M+1:T-1}$ and then marginalizing them out:

$$P^*(v_T \mid v_{1:M}) = \sum_{v_{M+1:T-1}} P^*(v_{M+1:T} \mid v_{1:M}). \tag{51}$$

Following the causal order, and the causal diagram of the target domain, we can write $P^*(v_{M+1:T} \mid v_{1:M})$ as a product of conditionals on the parents:

$$P^*(v_T \mid v_{1:M}) = \sum_{v_{M+1:T-1}} \prod_{i=M+1}^{T} P^*(v_i \mid \mathbf{pa}_i^*). \tag{52}$$

To transport the above, we attempt a multi-cause problem instance at every position $i$: for $i \in \{M+1, \cdots, K\}$ if there exists a position $i' \in [T]$ and domain index $j' \in [K]$ such that $\Delta(i, *; i', j') = 0$,

$$P^*(V_i = y \mid \mathbf{Pa}_i^* = \mathbf{x}) = P^j(V_{i'} = y \mid \mathbf{Pa}_{i'}^j = \mathbf{x}). \tag{53}$$

This allows us to pool data from all position-domain pairs $i', j'$ such that $\Delta(i, *; i', j') = 0$, and use it to estimate $P^*(V_i = y \mid \mathbf{Pa}_i^* = \mathbf{x})$. In Algorithm 1 $\mathcal{J}_i$ denotes this subset of position-domain pairs. Next, based on the parents in each of the source domains, we pool the data corresponding to $\mathcal{J}_i$, namely $D_i^{\text{TR}}$, with the size of $|\mathcal{J}_i| \cdot N + n$; the $n$ term is due to the fact that $\Delta(i, *; i, *) = 0$ is guaranteed. Next, compute the ERM using $D_i^{\text{TR}}$ to obtain $\mu_{\text{TR}}^i := \hat{P}(y \mid \mathbf{x}; D_i^{\text{TR}})$. Finally, we compose all these predictors, and marginalize out the intermediate variables to achieve an estimation of $P^*(v_T \mid v_{1:M})$:

$$\mu_{\text{TR}}(v_T \mid v_{1:M}) = \sum_{v_{M+1:T-1}} \prod_{i=M+1}^{T} \mu_{\text{TR}}^i(Y = v_i \mid \mathbf{X} = \mathbf{pa}^i). \tag{54}$$

Next, we decompose the risk of $\mu_{\mathrm{TR}}$ in terms of the risk of the predictors different positions:

$$R_{P*}(\mu_{\mathrm{TR}}) = \mathbb{E}_{P*}[-\log \mu_{\mathrm{TR}}(v_T \mid v_{1:M})] \qquad (\ell(\mu; y, \mathbf{x}) = -\log \mu(y \mid \mathbf{x}))$$

$$(55)$$

$$= \mathbb{E}_{P*}\big[\mathbb{E}_{P*}[-\log \mu_{\mathrm{TR}}(v_T \mid v_{1:M}) \mid v_{M+1:T-1}]\big] \qquad \text{(law of iterated expectation.)}$$

$$(56)$$

$$= \mathbb{E}_{P*}\big[\mathbb{E}_{P*}[-\log \sum_{v_{M+1:T-1}} \prod_{i=M+1}^{T} \mu_{\mathrm{TR}}^i(v_i \mid \mathbf{pa}_i^*) \mid v_{M+1:T-1}]\big] \qquad \text{(intermediate variables.)}$$

$$(57)$$

$$\leqslant \mathbb{E}_{P*}\big[\mathbb{E}_{P*}[\sum_{v_{M+1:T-1}} -\log \prod_{i=M+1}^{T} \mu_{\mathrm{TR}}^i(v_i \mid \mathbf{pa}_i^*) \mid v_{M+1:T-1}]\big] \qquad \text{(concavity of log \& Jensen ineq.)}$$

$$(58)$$

$$= \mathbb{E}_{P*}\big[\sum_{v_{M+1:T-1}} \sum_{i=M+1}^{T} \mathbb{E}_{P*}[-\log \mu_{\mathrm{TR}}^i(v_i \mid \mathbf{pa}_i^*) \mid v_{M+1:T-1}]\big] \qquad \text{(linearity of expectation)}$$

$$(59)$$

$$= \sum_{i=M+1}^{T} \mathbb{E}_{P*}\big[\sum_{v_{M+1:i-1}} \mathbb{E}_{P*}[-\log \mu_{\mathrm{TR}}^i(v_i \mid \mathbf{pa}_i^*) \mid v_{M+1:i}]\big] \qquad \text{(Markovianity)}$$

$$(60)$$

$$= \sum_{i=M+1}^{T} \mathbb{E}_{P*}[-\log \mu_{\mathrm{TR}}^i(v_i \mid \mathbf{pa}_i^*)] \qquad \text{(marginalize interm. vars.)}$$

$$(61)$$

$$= \sum_{i=M+1}^{T} R_{P*}(\mu_{\mathrm{TR}}^i) \qquad \text{(risk of sub-task)}$$

$$(62)$$

Let $\mu_*^i := P^*(v_i \mid \mathbf{pa}_i^*)$, so that,

$$\mu_*(v_t \mid v_{1:M}) = \sum_{v_{M+1:T-1}} \prod_{i=M+1}^{T} \mu_*^i(v_i \mid \mathbf{pa}_i^*). \qquad (63)$$

Due to Lemma C.4 and Corollary C.3, we have the following risk bound for the predictor at position $i$:

$$R_{P*}(\mu_{\mathrm{TR}}^i) - R_{P*}(\mu_*^i) \leqslant \mathcal{O}\big(\frac{|\mathcal{V}|^{|\mathbf{Pa}_i^*|+1}}{|\mathcal{J}_i| \cdot \epsilon^2 \cdot N + \epsilon \cdot n}\big). \qquad (64)$$

Therefore, we have the bound,

$$R_{P*}(\mu_{\mathrm{TR}}) - R_{P*}(\mu_*) \leqslant \sum_{i=M+1}^{T} R_{P*}(\mu_{\mathrm{TR}}^i) - R_{P*}(\mu_*^i) \qquad (65)$$

$$= \mathcal{O}\big(\sum_{i=M+1}^{T} \frac{|\mathcal{V}|^{|\mathbf{Pa}_i^*|+1}}{|\mathcal{J}_i| \cdot \epsilon^2 \cdot N + \epsilon \cdot n}\big). \qquad (66)$$

Let,

$$\mathcal{I} = \{i \in [T] \text{ s.t. } \mathcal{J}_i \neq \varnothing\}, \qquad (67)$$

denote the positions for which $P^*(v_i \mid \mathbf{pa}_i^*)$ is transportable. Also, let $c = \max_{i \in [T]} |\mathbf{Pa}_i^*|$. We have,

$$R_{P*}(\mu_{\mathrm{TR}}) - R_{P*}(\mu_*) = \mathcal{O}\big(\frac{|\mathcal{I}| \cdot |\mathcal{V}|^{c+1}}{\epsilon^2 \cdot N} + \frac{(T - M - |\mathcal{I}|) \cdot |\mathcal{V}|^{c+1}}{\epsilon \cdot n}\big). \qquad (68)$$

The latter justifies the claim of Theorem 2.7. We discuss the different rates achievable above in Appendix E.

## C.5 Proof of Theorem 3.2

The proof follows the logic the proof of Theorem B.5 (Appendix C.2). The circuit-TR procedure (Algorithm 2) partitions the position-domain pairs $i, j$ into clusters. The pairs $(i, *)$ each fall into a cluster, and we use the pooled data corresponding to the position-domain pairs in that cluster to estimate $P^*(v_i \mid \mathbf{pa}_j^*)$. In particular, $\mathcal{S} = \{\mathcal{E}_l\}$ in Algorithm 3 denotes these clusters, and we iterate over all of them. Next, we consider all combination of causal diagrams for the source and target domains; this allows the circuit-TR procedure to match the scope of the parents across the domains. For each combination above that corresponds to a selection diagram (i.e., $\Delta, \{\mathcal{G}^j\}_{j \in [K] \cup \{*\}}$), we use Algorithm 2 as a subroutine (StrInf) to compute a an estimation of $P^*(v_T \mid v_{1:M})$.

Notably, for all possible structures encoded as $\Delta, \{\mathcal{G}^j\}_{j \in [K] \cup \{*\}}$, we have a candidate in $\mathcal{H}$. Thus, the rate achieved by the circuit-TR procedure is matched by the minimum risk in $\mathcal{H}$, i.e.,

$$R_{P*}\left(\tilde{\mu} \in \arg\min_{\mu \in \mathcal{H}} R_{P*}(\mu)\right) = \mathcal{O}(R_{P*}(\mu_{\mathrm{TR}})). \tag{69}$$

Computing $\tilde{\mu}$ is only possible with large target data, however, we can compute an empirical risk minimzer within $\mathcal{H}$ using held-out target data to achieve similar rates. Let,

$$\mu_{\mathrm{AD}} \in \arg\min_{\mu \in \mathcal{H}} \frac{1}{|D^*_{\mathcal{D}_{\mathrm{test}}}|} \cdot \sum_{x,y \in D^*_{\mathcal{D}_{\mathrm{test}}}} \ell(\mu; y, x). \tag{70}$$

In computing $\mu_{\mathrm{AD}}$ we used held-out target data $D^*_{\mathcal{D}_{\mathrm{test}}}$ of size $\frac{n}{2}$, thus, we have,

$$R_{P*}(\mu_{\mathrm{AD}}) - R_{P*}(\tilde{\mu}) = \mathcal{O}\left(\sqrt{\frac{\log |\mathcal{H}|}{n}}\right). \tag{71}$$

We can bound the size of $\mathcal{H}$ as,

$$|\mathcal{H}| \leqslant \underbrace{(KT)^{KT}}_{\text{different partitions } \mathcal{S}} \cdot \underbrace{((2T)!)^{T^{K+1}}}_{\text{causal diagrams for all domains}} \tag{72}$$

which gives,

$$\log |\mathcal{H}| \leqslant KT \cdot (\log K + \log T) + (K+1)T \cdot T \log T = \mathcal{O}(KT^3 \log T). \tag{73}$$

The latter justifies the claim of Theorem 3.2:

$$R_{P*}(\mu_{\mathrm{AD}}) = \mathcal{O}\left(R_{P*}(\mu_{\mathrm{TR}}) + \sqrt{\frac{K \cdot T^3 \log T}{n}}\right). \tag{74}$$

## C.6 Proof of Proposition 4.1

Define,

$$\mu_\theta(v_i \mid v_{1:i-1}; j) := \Psi_\theta(v_i \mid A^j_{\theta\,i,\cdot} \cdot v_{1:T}; \Phi_\theta(i, j)). \tag{75}$$

We can rewrite the objective of Equation (12) as,

$$\mathcal{L}(\theta) := \lambda \cdot \underbrace{\left(d_\theta + \sum_{j=1}^{K} \sum_{i,i'=1}^{T} A^j_{\theta\,i,i'}\right)}_{\text{penalty}} + \underbrace{\sum_{j=1}^{K} \sum_{i=1}^{T} \mathbb{E}_{P^j}\left[D_{D_{\mathrm{KL}}}\left(P^j(\cdot \mid V_{1:i-1}) \| \mu_\theta(\cdot \mid V_{1:i-1}; j)\right)\right]}_{\text{match with source distributions}}. \tag{76}$$

In words, the objective in Equation (12) ensures that the solution entails a distribution that matches the sources at all conditionals and all domains, while preferring parameters with smaller mechanism indicator range $d$ and fewer edges in the graphs encoded by $\{A^j\}$.

The score can be decomposed into $K$ objectives as follows:

$$\mathcal{L}(\theta) = \lambda \cdot d_\theta + \sum_{j=1}^{K} \mathcal{L}_j(\theta), \tag{77}$$

where,

$$\mathcal{L}_j(\theta) = \lambda \cdot \|A_\theta^j\| + \sum_{i=1}^{T} \mathbb{E}_{P^j}\big[D_{D_{\mathrm{KL}}}\big(P^j(\cdot \mid V_{1:i-1})\|\mu_\theta(\cdot \mid V_{1:i-1}; j))\big)\big]. \tag{78}$$

For small enough $\lambda > 0$, maximizing $\mathcal{L}_j(\theta)$ ensures that the parent matrix $A^j$ has a one entry at the position of the true parents in each row, since $P^j(\cdot \mid v_{1:i-1}) = P^j(\cdot \mid \mathbf{pa}_i^j)$. Also, the penalty $\lambda \cdot \|A^j\|$ ensures that no additional one entries are kept in the parent matrix, thus, Equation (10) would be satisfied. Recall that $d$ is the size of the range of the mechanism indicator mapping $\Phi : [T] \times [K] \to [d]$.

If $\Delta(i, j; i', j') = 0$, we would have $P^j(v_i \mid \mathbf{pa}_i^j) = P^{j'}(v_{i'} \mid \mathbf{pa}_{i'}^{j'})$. To satisfy Equation (9), the mechanism indicator $\Phi : [T] \times [K] \to [d]$ must map $(i, j)$ and $(i', j')$ to the same value in the range $[d]$, which makes $\mu_\theta(v_i \mid \mathbf{pa}_i^j; j) = \mu_\theta(v_{i'} \mid \mathbf{pa}_{i'}^{j'}; j)$. By minimizing $\lambda \cdot d$, we ensure that this happens, satisfying Equation (9). Finally, note that once Equations (9) and (10) are satisfied, minimizing the divergence between the true distribution $P^*(v_i \mid v_{1:i-1})$ and $\mu_\theta(v_i \mid v_{1:i-1}; j)$ occurs only when Equation (11) is satisfied.

### C.7 PROOF OF PROPOSITION 4.2

The parameters to be learned in fine-tuning stage are:

1. The target mechanism indicator $\Phi^* : [T] \to [d_{\theta^{\mathrm{src}}}]$.
2. The target parent matrix $A^* \in [0, 1]^{T \times T}$.
3. The target-only predictors $\mu_i^*(v_i \mid v_{1:i-1})$.
4. The transport indicators $s_1, ..., s_T \in [0, 1]$

Once the pretrained parameters $\theta^{\mathrm{src}}$ satisfy Equations (9) to (11), consider the following values for the parameters of fine-tuning stage: Let $A^*$ encode the true causal diagram $\mathcal{G}^*$, and for the transported conditionals $P^*(v_i \mid \mathbf{pa}_i^*)$ we set $s_i = 1$, and $\Phi^*(i) = \Phi(i', j')$ for some $(i', j')$ which satisfies $\Delta(i, *; i', j') = 0$. For the non-transportable conditionals, we set $s_i = 0$ to use $\mu_i^*(v_i \mid v_{1:i-1})$ that is trained using the target data $D_{\mathcal{D}}^*$ of size proportionate to $n$. Let $\tilde{\theta}$ encode the parameters for this assignment of the fine-tuning parameters. We have,

$$R_{P*}(\mu_{\mathrm{ft}}^{\tilde{\theta}}(v_T \mid v_{1:M}) = \mathcal{O}(R_{P*}(\mu_{\mathrm{TR}}(v_T \mid v_{1:M})), \tag{79}$$

Since this set of values for the parameters corresponds to the circuit-TR solution. We discretize the fine-tuning parameter space into a set $\mathcal{H}$ of points , and consider only binary parent matrices and binary transport indicators. We can ensure that $\tilde{\theta}$ lies on this grid, among

$$|\mathcal{H}| \leqslant \underbrace{T^2}_{\text{parent matrix}} \cdot \underbrace{(KT)^T}_{\text{target mech. ind.}} \cdot \underbrace{2^T}_{\text{TR indicator}} \tag{80}$$

We use the held-out target data to obtain the best of these candidates,

$$\theta^* \in \arg\min_{\theta \in \mathcal{H}} \frac{1}{|D_{\mathcal{D}_{\mathrm{test}}}^*|} \cdot \sum_{v_T, v_{1:M} \in D_{\mathcal{D}_{\mathrm{test}}}^*} -\log \mu_{\mathrm{ft}}^\theta(v_T \mid v_{1:T}) \tag{81}$$

Compared to the best in class parameter set $\tilde{\theta}$, we would have an excess risk bounded as,

$$R_{P*}(\mu_{\mathrm{ft}}^{\theta^*}) - R_{P*}(\mu_{\mathrm{ft}}^{\tilde{\theta}}) = \mathcal{O}(\sqrt{\frac{\log |\mathcal{H}|}{n}}) = \mathcal{O}(\sqrt{\frac{T \cdot (\log K + \log T)}{n}}). \tag{82}$$

This proves,

$$R_{P*}(\mu_{\mathrm{ft}}^{\theta^*}) = \mathcal{O}(R_{P*}(\mu_{\mathrm{TR}}) + \sqrt{\frac{T \cdot (\log K + \log T)}{n}}) \tag{83}$$

$$= \mathcal{O}(R_{P*}(\mu_{\mathrm{TR}}) + \sqrt{\frac{K \cdot T^3 \cdot \log T}{n}}) = \mathcal{O}(R_{P*}(\mu_{\mathrm{AD}})) \tag{84}$$

## D  CHALLENGES DUE TO UNOBSERVED CONFOUNDERS

In this section we discuss how presence of confounders raises challenges in structure-informed and agnostic DA. We show that even in a very simple case, despite having the same causal functions generating $Y$ from $X$ and the unobserved variables between a source and target, transport is impossible.

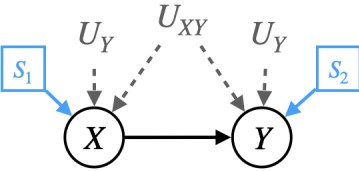

Figure 5: Selection diagram $\mathcal{G}^\Delta$

Let $X$ be a single binary covariate, and $Y$ be a binary label. Consider two source domains defined by the following SCMs:

$$\mathcal{M}^1 : \begin{cases} P^1(\mathbf{U}) : \begin{cases} U_X \sim \mathrm{Bern}(0.2) \\ U_Y \sim \mathrm{Bern}(0.05) \\ U_{XY} \sim \mathrm{Bern}(0.95) \end{cases} \\ \mathcal{F}^1 : \begin{cases} X \leftarrow U_X \oplus U_{XY} \\ Y \leftarrow (X \oplus U_{XY}) \oplus U_Y \end{cases} \end{cases} \qquad \mathcal{M}^2 : \begin{cases} P^2(\mathbf{U}) : \begin{cases} U_X \sim \mathrm{Bern}(0.9) \\ U_Y \sim \mathrm{Bern}(0.05) \\ U_{XY} \sim \mathrm{Bern}(0.95) \end{cases} \\ \mathcal{F}^2 : \begin{cases} X \leftarrow U_X \oplus U_{XY} \\ Y \leftarrow (X \oplus U_{XY}) \vee U_Y \end{cases} \end{cases}$$

Suppose the target domain is represented by the following SCM:

$$\mathcal{M}^* : \begin{cases} P^*(\mathbf{U}) : \begin{cases} U_X \sim \mathrm{Bern}(0.9) \\ U_Y \sim \mathrm{Bern}(0.05) \\ U_{XY} \sim \mathrm{Bern}(0.95) \end{cases} \\ \mathcal{F}^* : \begin{cases} X \leftarrow U_X \oplus U_{XY} \\ Y \leftarrow (X \oplus U_{XY}) \oplus U_Y \end{cases} \end{cases}$$

This setup induces the domain discrepancy sets $\Delta_{1,*} = \{X\}, \Delta_{2,*} = \{Y\}$; the corresponding selection diagram $\mathcal{G}^\Delta$ is shown in Figure 5. Although the mechanism of $Y$ is invariant between $\pi^*, \pi^1$, we can not transport $P^*(y \mid x)$; this follows from completeness results in transportability Pearl and Bareinboim (2011); Correa and Bareinboim (2020), that there exist source and target SCMs compatible with $\mathcal{G}^\Delta$, however, we would have $P^*(y \mid x) \neq P^1(y \mid x)$, e.g., $P^*(Y = 1 \mid x = 1) \approx 0.34$ but $P^1(Y = 1 \mid x = 1) \approx 0.06$.

Nonetheless, one can derive a tight bound for $P^*(y \mid x)$ Balke and Pearl (1997); in particular, we can compute $l_{y|x}, u_{y|x}$ such that,

$$P^*(y \mid x) \in [l_{y|x}, u_{y|x}]. \tag{85}$$

Moreover, this bound is tight, in the sense that for every $q \in [l_{y|x}, u_{y|x}]$, there exist source and target SCMs that entail $P^1, P^2$ over $X, Y$, induce the selection diagram $\mathcal{G}^\Delta$, such that we have $P^*(y \mid x) = q$. Deriving these bounds is called partial-transportability, and is studied in Jalaldoust et al. (2024). Even without any target data, although $P^*(y \mid x)$ is not transportable, we can compute a subset of conditional distributions that contains it through partial-transportability methods. In particular, let

$$\mathcal{P}^* = \{P^{\mathcal{M}_0^*}(x, y) \text{ s.t. } \mathcal{M}_0^1, \mathcal{M}_0^2, \mathcal{M}_0^* \text{ entail } P^1(x, y), P^2(x, y), \text{ and induce } \mathcal{G}^\Delta\}. \tag{86}$$

By definition, $P^*(x, y) \in \mathcal{P}^*(y \mid x)$ holds under the structural assumptions encoded in $\mathcal{G}^\Delta$. Thus, even without any target data, we can achieve non-trivial risk by solving the following min-max problem:

$$\mu_{\mathrm{pTR}} := \underset{\mu:\mathcal{X}\to\mathrm{simplex}^{|\mathcal{Y}|}}{\arg\min} \max_{\tilde{P}\in\mathcal{P}^*} R_{\tilde{P}}(\mu). \tag{87}$$

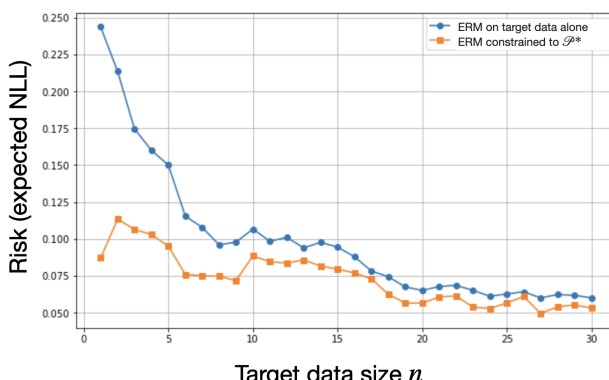

Figure 6: Risk of ERM on target data alone compared to ERM constrained to the partially transported set $\mathcal{P}^*$ constructed from $P^1(x,y), P^2(x,y), \mathcal{G}^\Delta$. As shown here, for smaller target data size there is a meaningful gap between the risk of the two estimators, but for larger $n$ this gap closes.

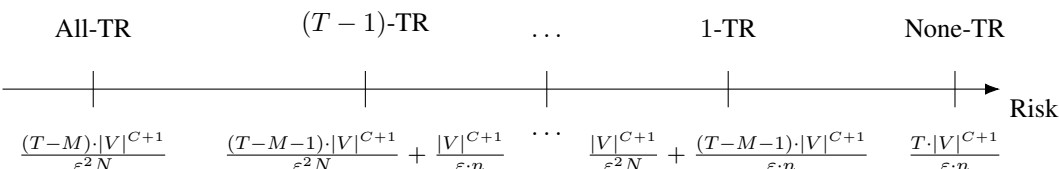

Figure 7: A schematic of risks obtained via the structure-informed procedure (Algorithm 2. In cases where all conditionals $P^*(v_i \mid v_{1:i-1})$ are transportable, we obtain a rate proportionate to $\frac{T-M}{\epsilon \cdot N}$. As more and more conditionals are non-transportable, they need to be estimated from the target data, adding a cost proportionate to $\frac{1}{n}$ for every non-transportable term. In the extreme case that no conditional is transportable, i.e., all target mechanisms are novel, we incur a risk proportionate to $\frac{T}{n}$.

A solution to this problem is called causal robust optimization (CRO) by Jalaldoust et al. (2024). Notably, with access to target data we can not significantly improve the risk achieved by $\mu_{\mathrm{pTR}}$; we elaborate on this point below.

Since $X, Y$ are binary, distributions over them can be parameterized by a point in $\mathrm{simplex}^3$. Consider $\mathcal{P}^*$ as a subset of the simplex, and suppose $P^*(x,y)$ lies in the *interior* of $\mathcal{P}^*$, that is, there exists $\delta > 0$ such that a $\delta$-ball around $P^*(x,y)$ is contained inside $\mathcal{P}^*$. Using vanilla ERM on the target data we can estimate the joint distribution $P^*(x,y)$, and form a confidence set that contains it. The diameter of this confidence set shrinks to zero by increasing the target data, due to uniform convergence properties of the ERM estimator. Take the target sample size $n_\delta$ such that the ERM confidence set containing $P^*(x,y)$ has a diameter smaller than $\delta$, and therefore, is contained entirely inside $\mathcal{P}^*$. Note that source data together with the structure $\mathcal{G}^\Delta$ only implies $P^*(x,y) \in \mathcal{P}^*$, so for target data of size $n > n_\delta$, the information $P^*(x,y) \in \mathcal{P}^*$ would be obsolete, since ERM on target data only achieves a strictly better confidence set for $P^*(x,y)$.

The above is unlike the unconfounded scenario; in case of transportability, the structure-informed procedure yields zero-shot generelization (i.e., rates in terms of the source data), that can not be beaten by vanilla ERM on the target data for any amount of target data.

To summarize, presence of confounders creates situations where certain quantities are partially transportable. This, in turn, enables non-trivial but imperfect generalization without any target data, and such situations can not occur without confounders. The immediate advantage of source data through partial transportability vanishes as the size of the target data grows, and this is shown in Figure 6.

□

# E    MORE DETAILS ON STRUCTURE-INFORMED RATES

In this section, we expand upon the possible rates in sequence adaptation via the structure-informed procedure. Please view the proof of Theorem 2.7 (Appendix C.4). We reduce the problem of estimating $P^*(v_T \mid v_{1:M})$ to estimating,

$$P^*(v_{M+1:T} \mid v_{1:M}) = \prod_{i=M+1}^{T} P^*(v_i \mid v_{1:i-1}). \tag{88}$$

Each of the conditionals, is either transported from a source domain (if there exists $(i', j')$ such that $\Delta(i, *; i', j') = 0$), or is estimated from the target data alone. In the former, the excess risk associated to estimation of $P^*(v_i \mid v_{1:i-1})$ would be $\frac{|\mathcal{V}|^{c+1}}{\epsilon^2 \cdot N}$, which is desirable since $N$ is large, and in the latter, the risk would be bounded by $\frac{|\mathcal{V}|^{c+1}}{\epsilon \cdot n}$. The joint risk depends on how many of the $T - M$ components are transported, and how many must be estimated from the target data. This gives a variety of rates, shown in Figure 7. Notably, if $N \gg n$, then we achieve a fast rate only if all components are transported, but achieve slower and slower rates for more and more non-transportable components. This figure is informative in the case of structure-agnostic adaptation as well, since due to Theorem 3.2 the risk of the structure-agnostic method is bounded by a fixed margin of $\sqrt{\frac{K \cdot T^3 \cdot \log T}{n}}$ compared to these rates, which is independent from the size of the vocabulary $\mathcal{V}$.

# F    SIMULATION SETUP AND REPRODUCIBILITY

The controlled simulation Section 4 involves a source domain and a target domain with sequences of length 10. To handle more than one parent, we follow the design discussed in Appendix G. We run simulation in a setting where each variable has at most two parents randomly selected from the previous variables in the causal order. Note that the causal diagrams of the source and target do not match necessarily, and are determined randomly and independently. The causal modules that determine the value of the variables are also drawn at random, from a pool containing null-ary, unary and binary noisy operators;

$$\mathcal{F} = \{g_{\mathrm{unif}}, g_{\mathrm{copy}}, g_{+1}, g_{-1}, g_{\times 2}, g_{\mathrm{sum}}, g_{\min}, g_{\mathrm{subtract}}, g_{\mathrm{mult}}\}. \tag{89}$$

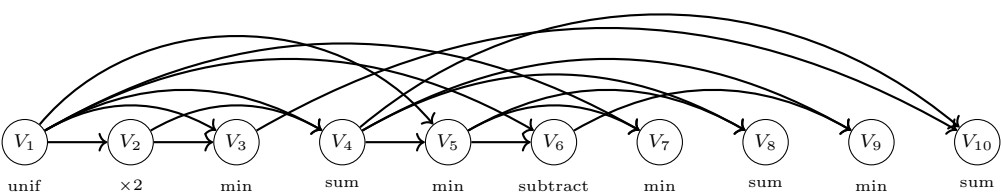

Figure 8: Causal diagram and operators corresponding to the source domain.

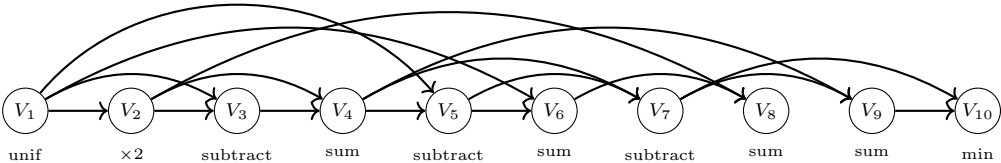

Figure 9: causal diagram and operators corresponding to the target domain.

The structure we investigated is shown in Figures 8 and 9, specifying the target and source SCMs, respectively. Next, we investigate pretraining and fine-tuning in these context of this SCMs.

### F.1 REPRODUCIBILITY

**Data generation** We evaluate our approach on a synthetic arithmetic benchmark where each sequence represents a functional program executed on base-10 digits.

**Model architecture** The `DomainAdaptationModel` uses the following specification:

- **Hidden dimension**: $r = 128$ (`config.hidden_dim`)
- **Positional encoding**: Learned embeddings of length $T$ and dimension $r$
- **Universal operator-indicator**: 2-layer MLP: $r \to d \to |\mathcal{F}|$ plus LayerNorm
- **Parent selector**: $H = 4$ causal attention heads with sharp-softmax temperature $\tau = 0.1$
- **Conditional MLP (token head)**: $2 \times (\text{Linear} + \text{ReLU}) + \text{LayerNorm}$, output size $|\mathcal{V}| = 10$
- **Maximum parents per head**: $C = 4$ (`config.max_parents`)
- **Parameter counts** (for $T = 20$): 43,868 total parameters, 20,946 trainable during fine-tuning
- **Activation dtype**: `float32` (no automatic mixed precision)

The **target-domain adapter** (our method) learns:

- New parent queries/keys (shape $[H, r, r]$ each)
- New operator-indicator MLP ($d \to d \to |\mathcal{F}|$)
- **Freezes**: positional encoding, base operator MLP, parent selector of source domain(s), and conditional MLP

**Training hyperparameters** Global optimizer: AdamW with $\beta_1 = 0.9$, $\beta_2 = 0.999$, $\varepsilon = 10^{-8}$, weight decay 0.01, batch size 32 sequences ($\to 32 \cdot (T - 1)$ tokens), no gradient clipping, constant learning rate schedule.

**Pre-training (source only)**:

- Epochs: 150
- Learning rate: $10^{-3}$
- Data: All source sequences (domain id 1)

**Fine-tuning (our method - adapter only)**:

- Epochs: 15
- Learning rate: $10^{-3}$
- Prefix length: $M = T/2$ (default, override with `-M`)
- Supervision modes:
  - **PS (process supervision)**: Full next-token cross-entropy
  - **NPS (no process supervision)**: Mask positions $\geqslant M - 1$

**Baseline configurations**:

- **ERM-Pooled**: Stage 1 (source only): 50 epochs, lr $10^{-3}$; Stage 2 (add target): 15 epochs, lr $5 \times 10^{-4}$, all parameters trainable
- **ERM-Joint**: Start from source pre-trained model, 20 epochs, lr $5 \times 10^{-4}$, all parameters trainable
- **Early stopping**: None

**Loss masking (NPS)**   In no-process-supervision (NPS) experiments, the cross-entropy at positions $\geqslant M - 1$ is excluded:

$$\text{mask}[i] = \begin{cases} 1 & \text{for } 0 \leqslant i < M - 1 \\ 0 & \text{for } i \geqslant M - 1 \end{cases} \tag{90}$$

Implemented in `build_loss_mask()` and applied on GPU.

**Evaluation protocol**

- **Input**: First $M$ tokens of target sequence where $M \in \{T - 1, T/2\}$
- **Generation**: Autoregressively sample positions $M, \ldots, T - 2$ by greedy argmax; keep logits for final position
- **Metric**: CE(logits$_{T-1}$, target digit) averaged over 1000 held-out target sequences
- **Seeds**: 3 independent runs (seeds $0, 1, 2$) with mean $\pm$ std reported in plots

**Computational environment**

- **Hardware**: NVIDIA H100 80GB (PCIe), CUDA 12.1, driver 535
- **Software**: PyTorch 2.1.0, Python 3.12, numpy 1.26
- **Determinism**: `torch.use_deterministic_algorithms(True)` and `torch.backends.cudnn.deterministic = True`
- **Typical runtimes**: $T = 10$, $N = 10^4$, $3 \times 3 \times 2$ grid $\approx 6$ minutes; $T = 20$, $N = 10^5$ pre-training $\approx 90$ minutes
- **Peak GPU memory**: $< 4$GB per worker

**Randomness control**   `random`, `numpy`, and `torch` are seeded at program start with deterministic offsets for DataLoader workers. Re-runs with identical seeds reproduce metrics within $10^{-8}$ precision. All experiments maintain deterministic seeding and full reproducibility through provided scripts and configurations.

## G   DETAILED MODEL ARCHITECTURE

This section provides a comprehensive walkthrough of our model architecture, focusing on how multiple parents are identified and utilized for next-token prediction in a domain-adaptive manner.

### OVERALL TASK AND CORE DESIGN PRINCIPLE

The objective is sequence modeling, specifically to predict the next token in a sequence of discrete symbols (digits 0–9 in our experiments). The core idea is that the generation of a token at position $i$ depends on:

1. A selected **causal function** (e.g., add, subtract, multiply)
2. One or more **parent tokens** from earlier positions in the sequence (i.e., positions $< i$)

### INPUT REPRESENTATION AND POSITIONAL ENCODING

**Input sequences**: Sequences of integer token IDs from $\mathcal{V} = \{0, 1, \ldots, 9\}$.

**Positional encoding** (`PositionalEncoding` class): Each token at position $(i, j)$ where $i$ is the sequence position and $j$ is the domain ID is mapped to a dense vector representation using:

- Standard sinusoidal positional encodings for both the position index ($0$ to $T - 1$) and the domain ID ($0$ to $K$)
- These two embeddings (each $r/2$ dimensional) are concatenated to form the initial $r$-dimensional embedding

- **Input**: $(B, T)$ for positions and domains; Output: $(B, T, d)$ embeddings, where $B$ is the batch size.

This design allows the model to be aware of both absolute position within the sequence and the domain context, enabling domain-specific parent selection as described in the following sections.

UNIVERSAL OPERATOR INDICATOR

The `UniversalOperatorIndicator` class determines, for each token position, the probability distribution over a fixed set of operations $\mathcal{F}$ (e.g., add, subtract, multiply_two).

**Mechanism**:

1. The $h$-dimensional embedding of each token is passed through a linear projection to $|\mathcal{F}|$ dimensions
2. Layer normalization is applied: LayerNorm(Linear(embedding))
3. Softmax produces a probability distribution: softmax(LayerNorm(Linear(embedding)))

**Universality**: This module's parameters are shared across all domains and positions. The *choice* of operation is contextual based on the token's embedding, but the *meaning* of each operation is universal across domains.

**Output**: $(B, T, |\mathcal{F}|)$ operator probabilities where $B$ is batch size.

DOMAIN-SPECIFIC PARENT SELECTOR

The `DomainSpecificParentSelector` class implements the core mechanism for identifying influential parent tokens, corresponding to the causal structure learning described in Algorithm 3.

**Multi-head causal attention design**: The mechanism uses $C$ distinct attention heads (where $C$ is the maximum number of parents). For our experiments, $C = 2$, meaning the model can identify up to two distinct parents for each token.

**Domain-specificity**: The key innovation is that query (Que) and key (Key) projection matrices are domain-specific:

$$\text{domain\_queries} \in \mathbb{R}^{(K+1) \times C \times r \times r} \tag{91}$$

$$\text{domain\_keys} \in \mathbb{R}^{(K+1) \times C \times r \times r} \tag{92}$$

During forward pass, for domain $j$ and parent head $h$:

$$\text{Que}_{j,h} = \text{Embeddings} \cdot W_{j,h}^{\text{Que}} \tag{93}$$

$$\text{Key}_{j,h} = \text{Embeddings} \cdot W_{j,h}^{\text{Key}} \tag{94}$$

**Parent selection process**: For each domain $j$, parent head $h$:

1. **Attention scores**: $S_{j,h} = \frac{\text{Que}_{j,h}\text{Key}_{j,h}^T}{\sqrt{r}}$
2. **Causal masking**: Standard causal attention masking ensures token at position $i$ only attends to positions $< i$
3. **Sharp softmax**: $A_{j,h} = \text{softmax}(S_{j,h}/\tau)$ where $\tau = 0.1$
4. **First position handling**: Weights for position 0 are zeroed as it has no parents

The temperature $\tau = 0.1$ makes the softmax significantly sharper, encouraging sparse selection of a small number of parents rather than soft averaging.

**Output**: For each domain $j$, a list of $C$ attention weight matrices $(B, T, T)$ representing parent selection distributions.

FEATURE PREPARATION FOR CONDITIONAL MLP

The `_prepare_mlp_features` method combines operator indicators and selected parent values to form input for the final prediction MLP.

**Input components**:

- sequences_onehot $\in \{0,1\}^{B \times T \times |\mathcal{V}|}$: One-hot encoded input sequences
- operator_indicators $\in [0,1]^{B \times T \times |\mathcal{F}|}$: From universal operator indicator
- parent_weights: From domain-specific parent selector
- domains $\in \{0,\ldots,K\}^{B \times T}$: Domain IDs

**Feature construction**: For each position $i$:

1. Operator indicators operator_indicators$[:,p,:]$ form the first part of the feature vector
2. For each parent head $h \in \{0,\ldots,C-1\}$:
$$\text{WeightedParentValue}_{h,p} = A_{j,h}[:,p,:] \cdot \text{sequences\_onehot} \tag{95}$$
   where $j$ is the domain of position $i$. This produces a $(B, |\mathcal{V}|)$ vector for each parent head.
3. These $C$ vectors are concatenated after the operator indicators

**Feature dimension**: $|\mathcal{F}| + (C \times |\mathcal{V}|)$ where $|\mathcal{F}|$ is the number of operations and $|\mathcal{V}| = 10$.

CONDITIONAL MLP FOR PREDICTION

The `EfficientConditionalMLP` class predicts the next token's probability distribution based on the combined features, implementing the learned conditional distributions $\hat{P}(v_i|\text{pa}_i)$ from our theoretical framework.

**Architecture**:

1. **Input projection**: Linear layer from feature dimension to $r$
2. **Hidden layers**: Stack of linear layers ($r \to r$) with ReLU activations, dropout, and residual connections
3. **Output layer**: Linear layer from $r$ to $|\mathcal{V}| = 10$

**Output**: Logits of shape $(B, T, |\mathcal{V}|)$ for next-token prediction.

TRAINING AND FINE-TUNING PROTOCOL

**Pre-training (source domains)**: The entire model, including domain-specific Que/Key matrices for source domains, is trained end-to-end using standard next-token prediction cross-entropy loss.

**Fine-tuning (target domain adaptation)**: When adapting to target domain $\pi^*$:

- **Frozen components**:
    - Positional encoding parameters
    - Universal operator indicator parameters
    - Conditional MLP parameters
    - Source domain Que/Key matrices
- **Trainable components**:
    - New randomly initialized $W_{*,h}^{\text{Que}}, W_{*,h}^{\text{Key}}$ for target domain
    - New target-specific operator indicator

This modular design enables learning domain-specific parent selection while reusing universal causal functions, corresponding to the structure-agnostic adaptation strategy in Algorithm 3 with computational efficiency discussed in Section 4.

