# OpenReview forum: "Adapting, Fast and Slow: Transportable Circuits for Few-Shot Learning"
_ICLR.cc/2026/Conference — Submitted to ICLR 2026_

### Official Review · Reviewer_jSKE · 2025-10-25

**Soundness:** 3
**Presentation:** 2
**Contribution:** 1
**Rating:** 2
**Confidence:** 3

**Summary:**

This paper studies the problem of domain adaptation (and domain generalization) from a perspective of causal structure transfer. Specifically, there are three main parts:
1. Transferring statistical information to a target domain when the causal structure of all domains is fully known and the target domain has a similar form to one of the source domains
2. Transferring information to a target domain when the target function can be constructed by composing the source domain functions, and details of each domain and their relationships are known
3. Using a small number of samples in the target domain to conduct an exhaustive search over a space of causal structures in order to conduct the procedure in part 2 when there is less domain information available
4. Approximating the exhaustive search with gradient-based optimization

**Strengths:**

1. The introduction does a good job of providing context, motivating the problem, and describing the contributions of the paper.
2. The theoretical results are relevant to the paper
3. The paper is generally clear about the assumptions made in each section
4. Corollary 3.5 is very interesting; perhaps this facet of the paper is worth exploring more.

**Weaknesses:**

1. The first methods require many assumptions to hold (in the form of extremely specific domain knowledge). While the method in section 3 resolves most of this, it is computationally intractable and it still requires the user to somehow determine $T^*$. The experiments with the gradient-based approximation are useful for resolving the first issue, but they are not entirely convincing; more on this in the next point.
2. The experiments could use several improvements:
    - Timing results, to see the cost of the proposed method compared to the baselines
    - More baselines (specifically, one that trains solely on the target data)
    - More tests in settings with varying amount of source-target causal transferability
    - More runs of each experiment, to verify reliability of the results
    - More explanation of the result in Figure 4b, where the method in the paper is significantly outperformed by the baseline (and doesn't seem to display any learning at all)
3. The writing (and math notation) got continually harder to follow throughout the paper. While the brief sentences to explain various formulas/algorithms are appreciated, the formulas and algorithms themselves are often very difficult to understand (especially in section 4). I suspect that the issue is that there is simply far too much notation going on (and many terms have both superscripts and subscripts); this gets difficult to parse very quickly, especially when different indices on a single symbol range over different types of objects.
4. The module-TR and circuit-TR algorithms seem trivial. The basic idea is to simply search over the possible causal "similarities" between the source domains and the target domain, identify and restructure the relevant data, and train the model. However, although the idea is simple, the algorithms and description are highly complex; it would take me a very, very long time to get this concept out of the pseudocode in algorithms 1 and 2. Simplicity and clarity are important goals in exposition.
5. Putting together equations (6) and (7), and comparing to equation (3) in the case where transport is not possible, makes it unclear whether the proposed circuit-AD method would ever actually perform well in a practical scenario; the additional term introduced by (7) seems like it would typically be very large, and it only scales with $n$ (target domain data) rather than $N$ (source domain data). The paper would significantly benefit from a discussion of this topic: under what realistic conditions is circuit-AD expected to perform better than simply training on the target domain data? Note that, in the small synthetic example 2.4, the value of $T^*=3|\mathcal{V}|$ would already be very large.
    - Relatedly, based on this, it seems that the sentence in the introduction (point 2, adaptation, on page 2) that gives the error rate is not correct; this error rate is in addition to the base error rate of circuit-TR, but that detail is not mentioned in the introduction.
    - The sentence "[...] since $n = \Omega(|\mathcal{V}|^3)$ is needed for a constant excess risk; in fact, the regular ERM on target yields similar error rate" seems to indicate that circuit-AD is, in fact, generally not better than simply training on the target data.

**Questions:**

1. On page 4, it is mentioned again that there is a positivity assumption, which (in the context of the sentence I am referring to) means that any function the system is trying to learn must be noisy. What are the ramifications of this requirement?
2. Section 4 mentions, in passing, another paper (Nichani et al. 2024) that studies the way that transformers learn causal structure. If existing architectures automatically learn causal structures, why is the gradient-based method proposed in the current paper necessary?

---

> ### Author Response · Authors · 2025-11-26
>
> Thank you for reviewing our work.
>
> > W1. Does the user need to determine $T_*$?
>
> Yes, for a fixed target sample size $n$ the user can choose $T_*$ to match their tolerance of excess risk, as outlines in Thm. 3.2; it is more likely that there exists a transport formula for larger $T_*$, which in turn enables *adaptation* at the cost of greater excess risk. This analogous to expanding the hypothesis class in iid setting to possibly gain from realizability.
>
> > W2.1. more experiments, and target-only ERM.
>
> Thank you for raising this, we will including more experiments with varying sequence length, vocabulary size, and circuit diversity to support the claims of the paper more strongly. Notably, the target-only ERM performance is reflected in ERM-joint, since in this baseline the target data is separated by a one-hot index, and doesn't interfere with sources.
>
> > W2.2. Explain Fig 4b where the method is outperformed by the baselines.
>
> The performance in Fig.4b is actually predicted by Theorem 3.2 as it is by design a case of slow adaptation; the circuit is not transportable with $T_* = M+1$, theory doesn't permit transfer. Thus, all methods need much more data to converge to the optimal risk of ~$0.5$.
>
> > W3. Complex notation.
>
> Do you have concrete recommendations to make the content more accessible? We had iterated many times over the results before arriving at this notation, and we find current presentation the most coherent while staying faithful to the content.
>
> > W4. The module-TR and circuit-TR algorithms seem trivial
>
> We respectfully disagree with the triviality sentiment. We acknowledge that learning complex tasks from a composition of simpler ones is a longstanding idea, but formalizing a fundamental/"simple" idea might seem easier said than done. We build on the causal transportability literature, and we hoped that remaining faithful to the original notation helps understanding the trade-offs and limitations, e.g., challenges of confounding.
>
> > W5.1. Comparing circuit-AD rates with target-only ERM.
>
> Theorem 3.2. guarantees the risk of circuit-AD as a *base risk* of circuit-TR with and an *excess* for not knowing the precise causal structure. In case of transportability, the base risk is equal to the **irreducible error** and all our claims are meant to refer to the excess risk w.r.t. this irreducible error, following the standard in the literature. The excess is $\sqrt{\frac{poly(K,T_*)}{n}}$, where $T_*$ is the length of the target sequence (that may be determined by the learner), $K$ is the number of sources, and $n$ is the target sample size. **Notably, the numerator is NOT the mere dimension of the task, as it does not necessarily grow with the vocabulary size $|\mathcal{V}|$, whereas the target-only ERM inevitably depends on it unless restrictive assumptions (e.g., linearity) are asserted.**
>
>
> > W 5.2. it seems [the error rate in point 2 on page 2] is not correct;
>
> Great catch; the word "excess" must precede "error".
>
> > W 5.3. [Line 348] seems to indicate that circuit-AD is not better than target-only ERM.
>
> Example 3.3. is indeed an example of slow adaptation, and in Example 3.4. we also discuss fast rates for the same task. Circuit transportability depends on the choice of $T_*$ *and* available source modules. Thus, circuit-AD can be slow/fast as we illustrated. In Ex 3.3. $T_* \in \Omega(|\mathcal{V}|)$ is needed for transport, however, in Ex. 3.4. by swapping $\mathrm{subtract}$ with $\mathrm{mod}$, transport is possible with $T_* \in O(\log |\mathcal{V}|)$, which in turn reduces excess risk achievable by circuit-AD. In general, circuit-AD is *very fast* when the target circuits length (composed of the source modules) does not depend on $|\mathcal{V}|$ at all, cf. (W2) in our response to reviewer Edck.
>
> > Q1. Limitations of positivity?
>
> Please see (W2) in response to Edck.
>
> > Q2. If Nichani et al. (2024) shows that 2-layer transformers learn causal graphs, why is your method necessary?
>
> The existing results don't apply since the architecture in Sec 4 is a modified 1-layer transformer, not considered by the reference above. Notably, "learning the graph" was never the objective in this work; it is merely a consequence of pre-training and is presented as such in the paper.
>
> **Finally, we would like to emphasize the following:**
>
> The paper's goal is not to propose a new *method*; it is a theoretical contribution to learning under distribution shifts with implicit causal "inductive biases". Alg 3. and Thm 3.2 are the main messages of the paper, and we firmly believe they are significant contributions to the field. The empirical demonstrations are not meant to make any claim beyond supporting this main message. The modified 1-layer transformer in Sec 4 is not intended to compete with SOTA few-shot learning methods, but to provably approximate the exhaustive search in Alg 3.
>
> **We hope that we have addressed all of the reviewer's concerns. Please let us know if we can clarify/elaborate further.**

---

> > ### Comment · Reviewer_jSKE · 2025-11-26
> >
> > Thank you for the thorough response. Since it has been about a month since my initial reading of the paper, I will re-read it with these points in mind. I will likely have additional comments sometime in the next few days.

---

### Official Review · Reviewer_Edck · 2025-10-31

**Soundness:** 3
**Presentation:** 3
**Contribution:** 3
**Rating:** 6
**Confidence:** 3

**Summary:**

This paper extends causal transportability theory to address zero-shot and few-shot domain adaptation. The authors introduce module-transportability and circuit-transportability defining formal criteria for when modules (local causal mechanisms) or compositions of such modules into higher-level circuits can be transferred from source to target domains. The authors present two algorithms: 1. Circuit-TR, which computes predictors for the target domain by composing transportable modules given explicit causal structure and a discrepancy oracle, 2. Circuit-AD that performs supervised domain adaptation without the causal structure by generating multiple candidate circuits and choosing the one that best fits a small amount of target data. The authors show synthetic experiments consistent with the theory.

**Strengths:**

- The link between causal transportability theory and circuit complexity is novel. The authors map few-shot adaptation rates to circuit size complexity and reframe sample-efficiency in terms of structural complexity.
- The work is theoretically sound: the authors prove strong control of excess risk for both structure-informed (Circuit-TR, Theorem 2.7) agnostic and agnostic adaptation (Circuit-AD, Theorem 3.2).
- The proposed gradient-based surrogate for Circuit-AD borrows attention-like components and is computationally viable. The demonstration in Fig.3 that the pre-training phase implicitly learns causal adjacency and discrepancy indicators is a particularly convincing empirical validation.

**Weaknesses:**

- Empirical validation is only on synthetic arithmetic sequences with a small number of observed variables ($T = 10$) and a small vocabulary size.  The gap between these experiments and real domain adaptation challenges is huge. While real-world experiments are not necessary, more empirical evaluation on synthetic sequences of varying $T$ and $\mathcal{V}$ would help understand the computational viability of the proposed algorithms, especially since Circuit-AD is exponential in $T$ (if I understood correctly).
- Can you offer some insights into how relying on known or perfectly recoverable causal graphs and strict positivity under discrete finite vocabularies translates into practice-- exactly what kind of datasets do these admit? What kind of real-world domain adaption data would it be possible to run the proposed algorithms on?
- In case of a failure of full transport (Appendix D), under what conditions could partial transport still yield some estimable/identifiable quantities?
- It would help to position this paper within the broader landscape of modular and compositional learning work  to understand its contributions beyond transportability.

I am happy to increase the score if the authors can address my concerns and questions.

**Questions:**

- Does the gradient-based surrogate provably approximate Circuit-AD in the limit of infinite data?
- The analysis is discrete and fully-observed. Can the framework extend to continuous or partially-observed systems using, e.g., additive noise models or variational causal discovery?
- The paper's setup of learning from multiple source domains to quickly adapt to targets resembles meta-learning. Could you add some explanation on the relationship between circuit transportability and gradient-based meta-learning algorithms like MAML?

---

> ### Author Response · Authors · 2025-11-26
>
> Thank you for your positive evaluation of our work.
>
> > W1. Empirical evaluation. [...] While real-world experiments are not necessary, more empirical evaluation on synthetic sequences of varying $T$ and $\mathcal{V}$ would help [...] especially since Circuit-AD is exponential in $T$ (if I understood correctly).
>
> We appreciate the comment on additional experiments, and commit to including them in the final manuscript. We see the need to point out a possible misunderstanding: Circuit-AD's sample complexity is NOT exponential in $T$; the price of not having access to the explicit structure is an excess risk of $\sqrt{\frac{KT^3}{n}}$ on top of what is achievable via circuit-TR (the graph-based algorithm), where $T$ is the length of the target sequence, $K$ is the number of source domains, and $n$ is the number of target samples. Importantly, this rate does not depend on the size of the vocabulary, as it does with target ERM. Also, our approach does not rely on any functional assumptions, e.g., linearity.
>
> > W2. what are the implications of positivity and the ground-truth causal structure in real-world DA tasks?
>
> **On positivity assumptions.** It is prevalent throughout causal inference and is critical for soundness, e.g., do-calculus is only valid under positivity. Intuitively, we avoid the collinearity-like situations such as the following: The SCM $V_2 \gets 10 \cdot V_1; V_3 \gets \frac{1}{2} V_2$ induces the graph $V_1 \to V_2 \to V_3$ and the observationally equivalent SCM $V_2 \gets 10 \cdot V_1; V_3 \gets 5 V_1$ induces a different $V_2 \gets V_1 \to V_3$, making the graph undiscoverable even with full observability.
>
> **On ground-truth graphical structure.** Any data where the label $Y$ follows a "noisy computation" applied to covariates $\mathbf{X}$ without confounding admits the structure discussed in this paper. For instance, a corollary to Theorem 3.2. is that it is possible to learn multiplication of matrices of size $M$ with *few* samples, once we have access large data from multip. of matrices of size $<M$. Intuitively, akin to GCD in Example 3.1, the reason for fast adaptation is that there *exists* a small circuit for $M$-sized multip. composed of sub-circuits $<M$-sized multip. **We emphasize that our goal is not to provide a new algorithm/architecture to compete with the SOTA few-shot learning; instead, our theory shows viability of compositional generalization using implicit causal structure, and Sec. 4 serves only as a proof of concept.**
>
> > W3. In case of a failure of full transport (Appendix D), under what conditions could partial transport still yield some estimable/identifiable quantities?
>
> We showed in Appendix D that the gains in few-shot rates rely heavily on absence of unobserved confounding.Partial-TR offers a *dense uncertainty set* for the quantity of interest using the source data, and this is only helpful for $n<n_0$ since for larger target sample size the empirical uncertainty set would lie inside of the partial-TR solution and deems it useless. In contrast, transportability even in presence of confounding can offer sample complexity gains compared to target-only ERM for all numbers of target data $n$, as outlined in Theorem 3.2.
>
> > W4. It would help to position this paper within the broader landscape of modular and compositional learning work to understand its contributions beyond transportability.
>
> We appreciate the comment and make sure to include a dedicated appendix to compositional learning and compare the results in detail. To our knowledge, the gains of compositional learning in few-shot settings are unexplored, though similar ideas exist in broader OOD context.
>
> > Q1. Does the gradient-based surrogate provably approximate Circuit-AD in the limit of infinite data?
>
> Yes; Proposition 4.2 shows this. The guarantee is indeed contingent on gradient descent finding/approximating the global optima, as it is difficult to make absolute guarantees with the non-convex optimization landscape.
>
> > Q2. Can we have extension to continuous variables, e.g. via ANM assumption?
>
> Great question! Yes, additional assumptions such as ANM fit nicely into the framework, as we can leverage them directly in discovering and estimating the modules $P(V_i \mid Pa_i)$, and this would enhance the sample complexity as well. With unobserved confounding, however, unless further assumptions are imposed Appendix D holds true, i.e., no provable gains.
>
> > Q3. [...] compare Circuit-TR and MAML.
>
> MAML aims to stays "a few gradient steps" away from the target domain's parameter before touching the target data, and achieves this by treating each source domain as a *mock* target domain in relation to others. Circuit-AD doesn't take such inductive leap, e.g., in Ex. 3.1. GCD is few-shot learnable via circuit-AD whereas none of the source circuits is few-shot learnable from the others.
>
> **We hope that we have addressed all of the reviewer's concerns; please let us know if we can clarify/elaborate further.**

---

### Official Review · Reviewer_XmEB · 2025-11-01

**Soundness:** 1
**Presentation:** 1
**Contribution:** 1
**Rating:** 2
**Confidence:** 4

**Summary:**

The draft is attempting to model the few-shot learning, or domain adaption, using ``causal'' graph as without assumption on the structures of variables in the problem we can not achieve transferable machine learning models. The main issue of the draft is that it is very likely the results are not based on causal graph, and instead, it is still based on correlation. The reason is that let's consider a causal system where X is observed data and Y is the label, the contents in X, may not be simply considered as the parent of Y and they can often be child if Y in a causal graph. Furthermore, even if X only contains the parent of Y, it may happen some contents of X are also the child of those part that are parents of Y, in this case, in general it may not be possible of recover the parent of Y from X without further assumption.

Furthermore, even if for the example case in Figure 2, if we only have X and Y, if may not be possible to recover V3, V4 and V5 from the observed X and Y. In this case, I believe the paper need a major improvement before publication.

**Strengths:**

There are just too many mistakes in the draft and thus I did not see any strengths of the paper.

**Weaknesses:**

1. Some of the theory part is wrong.

2. The claim of ``causal'' graph is wrong.

3. The experiment is weak and only simple simulation is considered and from the current model presentation I can hardly believe that the proposed model would work for a real-world setting.

**Questions:**

1. How do you justify the graph structure in the draft? I can hardly believe that in a real-world setting a graph would have similar structure.
2. Are all the V variables observable? If they are, then I believe classic approaches such as Lingam would work, if not, I believe the model is not identifiable in general.

---

> ### Author Response · Authors · 2025-11-26
>
> **The reviewer provided absolutely no reason for their negative evaluation. As we will discuss next, the summary section of the review does not carry any substance in relation to the content of our work. We believe the summary must be viewed as a proxy for credibility of the evaluation that follows it. We respectfully ask the AC/SAC to scrutinize the situation.**
>
> Below is a short description of the work for future readers:
>
> In section 2 *circuit transportability* scheme (Alg. 2) allows zero-shot generalization to the unseen target domain via explicit access to *causal structure* — that is a collection of causal graphs for each domain and the domain discrepancies across the domains. In Section 3, we show that if a target circuit is deemed transportable w.r.t. the ground truth causal structure, then even when the learner doesn't know the causal structure, they can still learn the target circuit in *few shots*, i.e., with only a *small* number of samples. The latter is illustrated via the *circuit adaptation* scheme (Alg. 3), and Theorem 3.2 establishes the precise rate, which is an excess over what is achievable through circuit-TR. This excess is $\sqrt{\frac{KT^3}{n}}$, growing polynomially in sequence size $T$ and number of source domains $K$, decaying with target sample size $n$. Importantly, this excess is independent of the vocabulary size, and doesn't rely on the functional forms, e.g., linearity.
>
> **Next, we iterate over the reviewer's summary, and continue with the weaknesses and questions.**
>
> >S1. The main issue of the draft is that it is very likely the results are not based on causal graph, and instead, it is still based on correlation. The reason is that let's consider a causal system where X is observed data and Y is the label, the contents in X, may not be simply considered as the parent of Y and they can often be child if Y in a causal graph.
>
> In lines 230-237 of the manuscript we make it explicit that $Y$ is assumed to be the last variable in the causal order, and as outlined throughout the paper, this does not trivialize the problem as it may be perceived by the reviewer.
>
> >S2. Furthermore, even if X only contains the parent of Y, it may happen some contents of X are also the child of those part that are parents of Y, in this case, in general it may not be possible of recover the parent of Y from X without further assumption.
>
> **This is a false statement.** In a Markovian graph (lines 230-237) $Y$ conditioned on its parents $Pa_Y$ is independent of ALL non-descendants, including the *siblings* mentioned by the reviewer.
>
> >S3. Furthermore, even if for the example case in Figure 2, if we only have X and Y, if may not be possible to recover V3, V4 and V5 from the observed X and Y. In this case, I believe the paper need a major improvement before publication.
>
> **This is a false statement.** Lines 223-226: In Example 2.4, the learner uses source data to learn the noisy $\min,\max,\mathrm{subtract}$ operators from the source domains. Then, by following the target graph and the domain discrepancies, the learner can compose
> $P^*(V_3,V_4,V_5\mid V_1,V_2) \approx \hat{P}^{\max}(V_3\mid V_1,V_2) \cdot \hat{P}^{\min}(V_4 \mid V_1,V_2) \cdot \hat{P}^{\mathrm{subtract}}(V_5\mid V_3,V_4)$, and similarly, it can compose the full noisy GCD operator.
>
> > W1. Some of the theory part is wrong.
>
> Which part?
>
> > W2. The claim of ``causal'' graph is wrong.
>
> The claims in this paper are 2.3,2.7,3.2,4.1,4.2; which one is "the claim of causal graph"?
>
> > W3. The experiment is weak and only simple simulation is considered and from the current model presentation I can hardly believe that the proposed model would work for a real-world setting.
>
> This work contributes to a theoretical literature, and the controlled experiments are designed to support the main message of the paper.
>
> > Q1. How do you justify the graph structure in the draft? I can hardly believe that in a real-world setting a graph would have similar structure.
>
> Any data that has an underlying computation graph with no confounding admits the structure, e.g., deciding the size of the largest clique in an $M$-node network; A corollary to Thm 3.2 is that learning this task via examples of $<M$ nodes enables few-shot learning to the $M$ node instances. We will add an appendix to elaborate on such examples further.
>
> > Q2. Are all the V variables observable? If they are, then I believe classic approaches such as Lingam would work, if not, I believe the model is not identifiable in general.
>
> Lline 234: $Y=V_{T}$ and $\mathbf{X} = \{V_1,...,V_M\}$ are the observables. LiNGAM is a causal discovery method that is not remotely relevant here.
>
> **Given the deep misconceptions outlined above, it is understandable if the reviewer is unable to provide an accurate assessment of the work. However, as they asserted a 4/5 *confidence*, we see the need to advise them to use this opportunity and recalibrate their confidence score.**
>
> Please let us know if we can clarify further.

---

### Official Review · Reviewer_p1up · 2025-11-01

**Soundness:** 3
**Presentation:** 1
**Contribution:** 2
**Rating:** 4
**Confidence:** 3

**Summary:**

The work extends classical causal transportability theory to compositional settings and formalizes mechanism‑level reuse across domains: if local mechanisms are shared, one can transport them and compose a target circuit even when the target function wasn't used in any of the sources as long as its decomposition was. Algorithms in no-confounder settings are introduced depending on the complexity of the target domain mechanisms and availability of the oracle. Gurantees in excess risk are given in zero- and few-shot learning cases. Finally a NN-based method is proposed for estimating the causal graphs and indicator function of shared mechanisms, with gurantees attached to it.

**Strengths:**

1. The work appears to be technically solid
2. The motivation for studying compositionality at circuit/mechanism level is clear

**Weaknesses:**

1. Broader context is unclear: the paper does not convincingly show how the proposed framework could be useful in real applications or what kinds of problems would benefit from it. It reads primarily as a technically detailed but narrowly scoped work rather than a conceptual contribution
2. Empirical results are weak both in breadth and depth; they appear to be sanity checks as opposed to an in-depth stress-tests of the proposals
3. Missing related work: the idea of reusing and composing operators is well-established in modular/meta-learning, the contribution appears to be a formalization inside causal transport.

**Questions:**

1. Is there a chance of testing the proposals in more realistic scenarios?
2. How does the approach relate empirically to modular/meta-learning architectures that already reuse mechanisms? A concrete example would be [1]


[1] Parascandolo, Giambattista, Niki Kilbertus, Mateo Rojas-Carulla, and Bernhard Schölkopf. “Learning Independent Causal Mechanisms.” arXiv:1712.00961. Preprint, arXiv, September 8, 2018. [https://doi.org/10.48550/arXiv.1712.00961](https://doi.org/10.48550/arXiv.1712.00961).

---

> ### Author Response · Authors · 2025-11-26
>
> Thank you for reviewing our work. Below we address the reviewer's concerns.
>
> ### **W1. Broader context.**
> > ... It reads primarily as a technically detailed but narrowly scoped work rather than a conceptual contribution
>
> The paper is indeed a theoretical work, and the scope is necessarily "narrow". **Our results relies on very specific set of assumptions that can be read as narrow/specific/precise**; e.g., consider the seminal work of Mansour et al. (2008), a "conceptual contribution" recognized by the community where the guarantees only apply to a quite specific setting that are also matched by the experiments.
>
> ### **W2. Empirical results.**
> > They appear to be sanity checks as opposed to an in-depth stress-tests of the proposals
>
> **Please notice that the goal of the paper is NOT devising a new method for domain adaptation**. Instead, our work characterizes the settings where the causal structure, though unknown to the learner, can make fast adaptation possible, *in principle*. To show this, we proposed methods (the symbolic search and the gradient-based heuristic) as a proof of concept, and designed controlled experiments accordingly. The neural architecture in the paper that resembles a 1-layer transformer is obviously not supposed to improve the benchmarks or solve real-world problems; rather it must solidify the main message and claims of the work from an empirical perspective, and we hope that it is serving the paper in this way. We believe "real-world" experiment do not add value to the current manuscript, since even a good performance on such benchmarks is not an evidence that would support the main message of the paper. Alternatively, we are still working on more diverse controlled experiments with varying problem parameters, and we hope to report by the end of the rebuttal window.
>
> ### **W3. Related work.**
> > The idea of reusing and composing operators is well-established in modular/meta-learning.
>
> We discussed the related work to the best of our ability; **please leave references for us if you believe we are missing related work beyond the one discussed in Q2.**
>
> The *ideas/intuition* of generalization through composition of modules has been discussed for a long time, even predating machine learning entirely. **The manuscript acknowledges this, but we stand by the claim that our work is the first formal study of few-shot learnability via composition of modules.** Analogously, general ideas about causality date back to Aristotle's Physics and Metaphysics, though the closest thing to a proper mathematical formulation was only achieve less than a century ago.
>
>
> ### **Q1. Is there a chance of testing the proposals in more realistic scenarios?**
>
> Recall our proposal: If there exists an unknown causal structure to the DA task, the learner doesn't need to know the structure to leverage it for fast adaptation. The controlled experiments are precisely designed to support this claim. We reframe your question as the following:
> >Is there a chance to test if there exist such causal structure in more realistic DA tasks?
>
> This work does not make any claim about realistic scenarios, and we believe this is appropriate. As an example, consider Vapnik's theory of generalization which proves that ERM performs well on the target when the source and target data are drawn from the same *well-behaved* distribution, i.e., the i.i.d. assumption. Is there a chance of testing whether the iid assumption holds in more realistic scenarios? Arguably, the iid assumption never holds in any realistic scenario, and the theory doesn't make a claim about this either; *if the assumptions hold*, then ERM is guaranteed to perform well.
>
> ### **Q2. How does the approach relate empirically to modular/meta-learning architectures that already reuse mechanisms? e.g., Parascandolo et al. (2018)**
>
> In this work, the target distribution is assumed to be a mixture of "independent" transformations of the source distributions, and the method trains one "expert" to learn the inverse of each transformation, e.g., the transformations add noise and the experts de-noise. The only hint of compositionality is the following line from the paper:
>
> > we test on Omniglot letters transformed with three consecutive transformations (noise, up left translation, contrast inversion) by
> applying the corresponding experts previously trained on MNIST, and correctly recover the original letters.
>
> Authors claim that the experts generalize through compositions, evident by the experiments where they apply the denoising in sequence to a different dataset to reverse the effect of a sequence of known noising processes. In contrast, compositions in our work are not necessarily one after the akin to the noise/denoise process, and most importantly, our proposal relies on the learner adapting to **unknown** compositions.
>
> We hope that we have addressed all of the reviewers comments; please let us know if we can clarify further.

---

### Author Response · Authors · 2025-12-04
**Final Remarks by the Authors**

We thank the reviewers and area chairs for their engagement with our work. Below we provide a summary of the paper, followed by a summary of the reviews and our clarifications.

### **Summary**
In Section 2, Circuit-TR (Alg. 2) enables zero-shot generalization to unseen target domains via explicit access to *causal structure* —a collection of causal graphs and domain discrepancies that can be viewed as a *recipe* for the compositional/circuit structure in the target domain in relation to the source *modules*. In practice, this domain knowledge may be unavailable, so in Section 3 we develop Circuit-AD (Alg. 3), an algorithm that allows learning the target circuit in *few shots* in cases where the target circuit is in fact transportable but the learner have access to the true causal structure to invoke circuit-TR routine. Specifically, Theorem 3.2 establishes the sample complexity as an *excess risk* of $\sqrt{\frac{K^3\cdot T}{n}}$ over what is achievable via Circuit-TR, where $T$ is sequence length, $K$ is the number of source domains, and $n$ is target sample size. Crucially, this rate is independent of vocabulary size (unlike target-only ERM) and requires no functional assumptions (e.g., linearity). Circuit-AD is computationally intractable, so in Section 4 we provide a gradient-based surrogate that provably approximates Circuit-AD, and our controlled experiments validate the theoretical claims of the paper regarding surrogate matching the symbolic scheme, and the fast and slow adaptation regimes predicted by the theory.

### **Strengths acknowledged by the reviewers**
1. **Clear problem formulation and well-motivated contribution:** the introduction "does a good job of providing context, motivating the problem, and describing the contributions" (jSKE).
2. **Technically solid work:** Theorems 2.7 and 3.2 provide actionable bounds, not just asymptotic rates (p1up, Edck).
3. **Establishes a novel connection between causal inference and few-shot learning:** The work extends theory of transportability to account for compositional generalization, reframing sample-efficiency in terms of structural complexity, opening new research directions (Edck).
4. **Empirically validated gradient-based surrogate method:** Figure 3 shows pre-training implicitly recovers causal structure, a "particularly convincing empirical validation" (Edck).


### **Reviewers' concerns and authors' clarifications**
1. **Broader context and real-world applicability (p1up, Edck, jSKE).** We emphasize that our work is a theoretical contribution to an extensive literature on domain adaptation, and such work necessarily defines a precise scope. Our guarantees apply under specific assumptions as do seminal works like Mansour et al. (2009) and many other examples. To motivate real-world applicability of the framework, we emphasized that *any* data where the label $Y$ follows a noisy computation over covariates $X$ without confounding admits our assumptions. For instance, a corollary to Theorem 3.2 is that M by M matrix multiplication is few-shot learnable from data on smaller matrices.

2. **Empirical evaluation limited to synthetic settings (p1up, Edck, jSKE).**
The experiments are intentionally controlled/synthetic with known ground truth. Our goal is to characterize when causal structure enables fast adaptation *in principle*, not to propose a new benchmark method. The neural architecture proposed in Section 4 that resembles a one-layer transformer is merely a proof of concept and shall not be viewed as a solution to real-world few-shot learning tasks.

3. **Positioning within modular/meta-learning literature (p1up, Edck).**
We provide the first formal study of few-shot learnability via module composition with precise sample complexity guarantees. Unlike prior work (e.g., Parascandolo et al., 2018; MAML), our approach handles unknown compositions without committing to a notion of *symmetry* between the domain.

4. **When does Circuit-AD outperform target-only ERM (jSKE)?**
Circuit-AD's rate is independent of $|V|$, while target ERM scales with $|V|$. When $T_* \ll |V|$ (small transportable circuit exists), gains are substantial. Example 3.4 shows how swapping one of the source domains guarantees $T_* \in \mathcal{O}(\log |V|)$ is achievable, enabling fast adaptation.

5. **Regarding Reviewer XmEB's assessment.**
The content of review shows fundamental misunderstandings and wrong presumptions that we detailed in our rebuttal, and we respectfully request the AC/SAC to scrutinize validity of the evaluation that follows this review.

### **Conclusion.**
Our work is the first (to our knowledge) that translates compositional/causal structure of the tasks to sample complexity gains in domain adaptation, even in absence of an explicit domain knowledge. We believe that our work makes a foundational contribution to understanding when, why, and how *fast* adaptation is possible in the under explored data modality of sequential prediction and generation.

---

### Meta-Review · Area_Chair_Ay1x · 2025-12-28

**Summary:**

This submission received **highly mixed but overall low initial scores**, ranging from explicit rejects (Reviewer jSKE), through marginal rejects (Reviewer p1up), to one borderline accept (Reviewer Edck). There was no reviewer consensus in favor of acceptance during discussion phase. Reviewer XmEB did not provide sufficient reasoning to justify their evaluation, and therefore their comments were not considered in the final assessment.

In addition, reviewers acknowledge that the paper contains nontrivial theoretical contributions, but raise serious concerns about the practical and empirical relevance of the results and find the presentation overly complex and difficult to follow. Although the authors emphasize that the paper is primarily theoretical, providing more substantial empirical validation beyond simple simulations would help demonstrate scalability to larger systems and broaden the paper’s potential audience. Based on the initial scores and the unresolved nature of the core concerns, my judgment as AC is to **Reject**.

**Reviewer Concerns:**

**Concerns partially addressed by the rebuttal:**

* **Scope and intent as a theoretical contribution** were clarified, emphasizing that the work is not meant to propose a competitive algorithm but to characterize few-shot learnability under compositional causal structure
  *(Reviewer p1up, Reviewer Edck)*.

* **Relationship to modular/meta-learning** was discussed conceptually, with clearer distinctions between known-module composition and adaptation to unknown compositions
  *(Reviewer p1up, Reviewer Edck)*.


**Concerns that remain outstanding:**

* **Extremely narrow applicability and weak empirical support**:
  Multiple reviewers note that experiments are **limited to small, synthetic settings** and function mainly as sanity checks, with little evidence that the theory connects to realistic domain adaptation problems
  *(Reviewer p1up, Reviewer Edck, Reviewer jSKE)*.

* **Presentation and accessibility issues affecting evaluability**:
  Poor presentatio and dense notation make it difficult to assess even the theoretical contributions
  *(Reviewer p1up, Reviewer jSKE)*.

* **Unclear advantage over simpler baselines**:
  Questions remain about whether circuit-AD meaningfully improves over target-only ERM except in highly constrained cases, and whether claimed rates translate to practical gains
  *(Reviewer jSKE)*.

**Reviewer Scores:**

* **Reviewer jSKE (initial: 2)** → **Likely unchanged or slightly higher**
  Concerns about assumptions, practicality, and unclear benefit over ERM persist despite rebuttal.

* **Reviewer p1up (initial: 4)** → **Likely unchanged**
  Rebuttal clarifies intent but does not address lack of compelling empirical evidence or broader impact.

* **Reviewer Edck (initial: 6)** → **Possibly unchanged**
  Positive on theory, but empirical gap and scope limitations remain; unlikely to shift overall decision.

---

### Decision · Program_Chairs · 2026-01-26

Reject